# EFFICIENT FORWARD-ONLY DATA VALUATION FOR LLMS AND VLMS

## ABSTRACT

Quantifying the influence of individual training samples is essential for enhancing the transparency and accountability of large language models (LLMs) and vision-language models (VLMs). Existing data valuation methods rely on Hessian information or model retraining, making them computationally prohibitive for billion-parameter models. In this work, we introduce `For-Value`, a forward-only data valuation framework that enables scalable and efficient influence estimation for both LLMs and VLMs. By leveraging the rich representations of modern foundation models, `For-Value` computes influence scores using a simple closed-form expression based on a single forward pass, thereby eliminating the need for costly gradient computations. Our theoretical analysis demonstrates that `For-Value` accurately estimates per-sample influence by capturing alignment in hidden representations and prediction errors between training and valuation samples. Extensive experiments show that `For-Value` matches or outperforms gradient-based baselines in identifying impactful fine-tuning examples and effectively detecting mislabeled data.

## 1 INTRODUCTION

Modern large language models (LLMs) and vision-language models (VLMs) have achieved remarkable success across a wide range of applications, driven by the power of large-scale pretraining (Achiam et al., 2023). These pretrained models are subsequently fine-tuned for tasks such as machine translation, dialogue systems, medical diagnosis, and multimodal reasoning (Guo et al., 2025; Bai et al., 2025b; Wu et al., 2025; Shao et al., 2024; Hao et al., 2025). Despite their impressive performance, these models remain prone to generating factually incorrect or biased outputs (Deng et al., 2023; Ferrara, 2023), often due to the presence of irrelevant, mislabeled, or unrepresentative training data. This highlights the need for scalable methods to quantify the impact of each individual training data and select the high-value samples that benefit the targeted tasks.

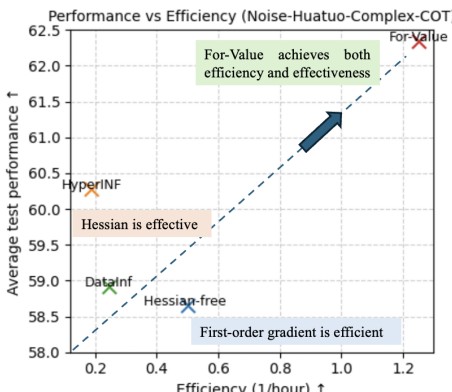

Figure 1: Comparison of data valuation methods in terms of effectiveness and efficiency when selecting training data from the Noise-Huatuo-Complex-CoT dataset for fine-tuning.

The data valuation task aims to assign scores to each training sample based on their effect on model performance on a valuation set (e.g., validation data) (Wang et al., 2024a), where performance is commonly assessed using loss, margin, or likelihood (Bae et al., 2024). Notable approaches include influence functions (Kwon et al., 2024) and Shapley value-based methods (Ghorbani & Zou, 2019), which provide frameworks for estimating how individual data points affect model predictions (Kwon et al., 2024; Zhou et al., 2024). These methods have proven effective in downstream applications such as detecting mislabeled data (Koh & Liang, 2017; Kwon et al., 2024), identifying influential examples, diagnosing bias (Kong et al., 2021), and auditing datasets (Grosse et al., 2023). However, influence function and Shapley value methods are computationally prohibitive for large models due to their reliance on Hessians and repeated retraining.

Figure 2: Pipeline of `For-Value`. Given a valuation sample and a training dataset, `For-Value` performs a forward pass over all data to compute scores (Eq. (1)) for each training example, using the last hidden embeddings and the prediction error $\alpha$. The training samples are then ranked based on these computed values.

To alleviate the high computational cost of influence estimation, several approximation techniques have been introduced. TracIn (Pruthi et al., 2020) estimates data influence by tracking gradient similarity across training checkpoints, while DataInf (Kwon et al., 2024) and HyperInf (Zhou et al., 2024) focus on efficient Hessian approximations. These methods, however, involve notable trade-offs: TracIn requires storing numerous model snapshots; DataInf suffers from approximation errors that scale with model size; and HyperInf assumes gradient independence and incurs cubic computational complexity. In parallel, for Shapley value approximation, Wang et al. (2024a) propose an online method that measures gradient or Hessian similarity between valuation and training data during training. However, applying this method to individual valuation samples remains impractical due to the need to compute and store per-sample gradients at every training step. Crucially, all these methods depend on access to model gradients and fine-tuned weights—resources that are often inaccessible in practical LLM and VLM deployments. Alternative strategies, such as similarity-based methods used in classification tasks (Just et al., 2023) and generative image models (Yang et al., 2025), are less applicable to LLMs and VLMs, as their foundational assumptions conflict with the training and inference processes of these models.

In this work, we introduce `For-Value`, a forward-only data valuation framework tailored for LLMs and VLMs. Instead of relying on gradients or model retraining, which are computationally expensive for evaluating data for LLMs and VLMs, we introduce a novel approach to analyzing the influence of training samples on the change in valuation data's likelihood, using only a forward pass. Specifically, we focus on the rich and informative hidden representation (Mixon et al., 2022; Deng et al., 2025; Zhao et al., 2024), and propose `For-Value` that unfolds the change in likelihood into a closed-form measure that captures both representation similarity and prediction error alignment between training and valuation samples. This alignment measure enables `For-Value` to identify influential or mislabeled data using only a single forward pass, making it highly scalable and practical. As illustrated in Fig. 1, `For-Value` achieves both effectiveness and superior efficiency compared to prior data valuation methods. Our contributions are as follows:

• We propose `For-Value`, a forward-only framework for identifying influential or noisy training data when adapting pretrained LLMs and VLMs to downstream tasks.

• We establish a theoretical foundation showing that, under the standard next-token prediction objective (token-level cross-entropy), data valuation can be approximated by the alignment between hidden representations and prediction errors.

• We show empirically that `For-Value` reliably equals or exceeds baseline performance in detecting influential and mislabeled samples, enhances downstream fine-tuning, and achieves these gains with vastly superior efficiency.

## 2 RELATED WORK

**Pretrained LLMs and VLMs.** In modern machine learning workflows, it is standard practice to utilize pretrained foundation models (FMs) and adapt them to specific downstream tasks (Deng et al., 2024a; Dettmers et al., 2023). Foundation models, such as large language models and vision-language models, serve as powerful initialization points thanks to their extensive pretraining on large-scale datasets. LLMs, including LLaMA (Touvron et al., 2023) and GPT-4 (Achiam et al.,

2023), are trained on diverse textual data for language understanding and generation. VLMs, such as Qwen2.5-VL (Bai et al., 2025a), LLaMA-VL (Meta, 2024), and GPT-4V (Yang et al., 2023), integrate visual and textual inputs to perform tasks like image captioning and visual question answering.

**Data Valuation.** The goal of data valuation is to quantify the contribution of each training example in $\mathcal{D}^{\text{train}}$ to the model's performance on a targeted valuation set $\mathcal{D}^{\text{val}}$ (e.g., validation data) (Wang et al., 2024a). With common metrics including loss, margin, or likelihood (Bae et al., 2024). Influence estimation is a widely adopted technique for quantifying the data value. The Hessian-based method introduced by Koh & Liang (2017) leverages second-order derivatives to compute influence functions but becomes computationally prohibitive for large-scale models. More recently, Bae et al. (2024) employed influence functions to evaluate data value across different training stages; however, the computational cost grows with the number of stages considered, making the method expensive in practice. To improve efficiency, methods such as DataInf (Kwon et al., 2024) and HyperInf (Zhou et al., 2024) propose efficient approximations that bypass explicit Hessian inversion. Nevertheless, all these influence function based methods require finetuning the model first. Similarly, TracIn (Pruthi et al., 2020) adopts a Hessian-free approach by tracking first-order gradients across training checkpoints to estimate data influence, but it requires storing and accessing many checkpoints, which is impractical for large models. Beyond influence-based methods, Shapley value based techniques (Ghorbani & Zou, 2019) assess data importance through marginal contributions. While theoretically appealing, these methods are computationally expensive due to the need for repeated model training. To mitigate this, Wang et al. (2024a) propose an online Shapley value approximation by measuring the similarity between valuation and training gradients during training. However, extending this approach to individual data points remains impractical, as it necessitates computing and storing per-sample gradients at every training step. In contrast to these methods, our approach neither requires finetuning the model nor backpropagation.

## 3 PRELIMINARIES

**Auto-Regressive Pretrained LLMs and VLMs.** We examine a pretrained large language model (LLM) or vision-language model (VLM) denoted as $\pi_\theta$, where $\theta$ represents its parameters. For a given input $\boldsymbol{x}$ — which may consist of text tokens, image patches, or a combination of both — the model defines a conditional probability distribution over an output text sequence $\boldsymbol{y} = (y_1, y_2, \ldots, y_{|\boldsymbol{y}|})$, factorized as:

$$\pi_\theta(\boldsymbol{y}|\boldsymbol{x}) = \prod_{k=1}^{|\boldsymbol{y}|} \pi_\theta(y_k|\boldsymbol{x}, \boldsymbol{y}_{<k}),$$

where $\boldsymbol{y}_{<k} = (y_1, \ldots, y_{k-1})$. At each step, the model predicts the next token $y_k$ conditioned on the input $\boldsymbol{x}$ and the prefix $\boldsymbol{y}_{<k}$. This auto-regressive structure underlies most modern LLMs and VLMs, which are used in tasks such as text generation (Wu et al., 2025), image captioning (Bai et al., 2025a), and multi-modal reasoning (Achiam et al., 2023).

## 4 METHOD

In this section, we provide a theoretical foundation for understanding how individual training examples influence the behavior of LLMs/VLMs on target valuation data. These insights motivate the design of our proposed method `For-Value`.

### 4.1 FORWARD-ONLY DATA VALUATION

**Notation:** Let $\boldsymbol{W}$, $\boldsymbol{w}_z$, and $\boldsymbol{h_z}$ denote the token unembedding matrix, unembedding of a token $z \in \mathcal{V}$, where $\mathcal{V}$ is the vocabulary, and hidden embedding of generated tokens $\boldsymbol{z} \in \mathcal{V}^*$ with embedding dimension $d$, respectively. Let $\boldsymbol{z}_k$ be the $k$-th token in $\boldsymbol{z}$ and $\boldsymbol{z}_{<k}$ be the first $k-1$ tokens in $\boldsymbol{z}$. Lastly, we denote by $\boldsymbol{e}_z \in \mathbb{R}^{|\mathcal{V}|}$ the standard basis vector corresponding to $z \in \mathcal{V}$.

Formally, given a training dataset $(\boldsymbol{x}_i, \boldsymbol{y}_i)_{i=1}^n \in \mathcal{D}^{\text{train}}$ and a valuation sample $(\boldsymbol{x}_v, \boldsymbol{y}_v) \in \mathcal{D}^{\text{val}}$, we define the notion of *Data Value* as follows:

**Definition 1** (Data Value). *At any training time $t > 0$, a training sample is considered more valuable to a given data point $(\boldsymbol{x}_v, \boldsymbol{y}_v)$ if it results in a greater likelihood change $\frac{d}{dt} \ln \pi_{\theta(t)}(\boldsymbol{y}_v|\boldsymbol{x}_v)$.*

This definition captures how much a training sample improves the model's confidence in predicting $(\boldsymbol{x}_v, \boldsymbol{y}_v)$. A higher likelihood corresponds to a lower loss on the valuation data during LLM/VLM fine-tuning. More broadly, our definition of data value is closely tied to the perplexity metric, which inversely reflects the model's uncertainty in text generation. We then analyze the learning dynamics of the valuation log-likelihood, $\frac{d}{dt} \ln \pi_{\theta(t)}(\boldsymbol{y}_v \mid \boldsymbol{x}_v)$, which characterizes the objective of increasing the probability of generating valuation outputs. In this work, we **focus on the pretrained model** ($t = 0$) and, for brevity, omit the time index $t$ in subsequent discussions. We begin with the following assumption:

**Assumption 1** (Unconstrained Features). *Expressive (enough) neural networks (e.g., pretrained LLMs/VLMs) can produce unconstrained embeddings $\mathbf{h}_{\boldsymbol{x}} \in \mathbb{R}^d$ independent of the architecture's specific complexities (Mixon et al., 2022; Deng et al., 2025; Zhao et al., 2024). These embeddings are subsequently transformed into logits by a token unembedding matrix $\mathbf{W} \in \mathbb{R}^{|\mathcal{V}| \times d}$. The resulting logits are passed through a softmax function to yield a probability distribution over possible next tokens. To assign probabilities to sequences $\boldsymbol{y} \in \mathcal{V}^*$, the language model $\pi_\theta$ operates in an autoregressive manner, i.e., $\pi_\theta(\boldsymbol{y} \mid \boldsymbol{x}) = \prod_{k=1}^{|\boldsymbol{y}|} \mathrm{Softmax}(\mathbf{W}\mathbf{h}_{\boldsymbol{x}, \boldsymbol{y}_{<k}})_{y_k}$.*

Notably, the unconstrained feature assumption has been widely adopted in the analysis of pretrained LLMs (Mixon et al., 2022; Razin et al., 2024; Zhao et al., 2024). For example, it has been leveraged in reinforcement learning studies (Deng et al., 2025; Razin et al., 2024) and in geometric analyses of LLM representations (Zhao et al., 2024), reinforcing its role as a foundation for `For-Value`. Under the unconstrained feature setting, the influence of a training sample on valuation sample is represented as (detailed proof in Appendix):

**Theorem 1.** *For a sample $\boldsymbol{x}_v$ and its generation $\boldsymbol{y}_v$ that await valuation, when fine-tuning a pretrained model using a training sample $(\boldsymbol{x}_i, \boldsymbol{y}_i), i \in [n]$, when no training input $\boldsymbol{x}_i$ is identical to the valuation input $\boldsymbol{x}_v$[1], the training data exhibits larger value to the valuation data as the following increases:*

$$\sum_{k=1}^{|\boldsymbol{y}_v|} \sum_{k'=1}^{|\boldsymbol{y}_i|} \alpha_{k,k'} \cdot \left\langle \boldsymbol{h}_{\boldsymbol{x}_v, \boldsymbol{y}_{v,<k}}, \boldsymbol{h}_{\boldsymbol{x}_i, \boldsymbol{y}_{i,<k'}} \right\rangle \tag{1}$$

*where $\alpha_{k,k'} = \left\langle \mathbf{e}_{\boldsymbol{y}_{v,k}} - \pi_\theta(\cdot \mid \boldsymbol{x}_v, \boldsymbol{y}_{v,<k}), \mathbf{e}_{\boldsymbol{y}_{i,k'}} - \pi_\theta(\cdot \mid \boldsymbol{x}_i, \boldsymbol{y}_{i,<k'}) \right\rangle$ quantifies the similarity of token-level prediction error across samples.*

As established in the theorem, the data value arises from the alignment between hidden representations and prediction errors (effect of prediction error see Sec. 6.3). A larger score of Eq. (1) indicates a greater increase in the likelihood of the valuation data, and hence a higher value. Since this score can be computed with only a single forward pass, we refer to Eq. (1) as `For-Value`.

## 4.2 Implementation of `For-Value`

Having introduced `For-Value`, we now describe its practical computation for scalable implementation. Fig. 2 illustrates the overall pipeline of our method, with further details provided below.
**Matrix Similarity:** First, we rewrite (1) into the form of a matrix inner product.

$$\left\langle \sum_{k=1}^{|\boldsymbol{y}_v|} \left( \mathbf{e}_{\boldsymbol{y}_{v,k}} - \pi_\theta(\cdot|\boldsymbol{x}, \boldsymbol{y}_{v,<k}) \right) \boldsymbol{h}_{\boldsymbol{x}_v, \boldsymbol{y}_{v,<k}}^T, \sum_{k'=1}^{|\boldsymbol{y}_i|} \left( \mathbf{e}_{\boldsymbol{y}_{i,k'}} - \pi_\theta(\cdot|\boldsymbol{x}, \boldsymbol{y}_{i,<k'}) \right) \boldsymbol{h}_{\boldsymbol{x}_i, \boldsymbol{y}_{i,<k'}}^T \right\rangle \tag{2}$$

Importantly, our reformulation involves calculating the summations over $k, k'$ before taking the inner product. This reformulation reduces the overall complexity to that of a single matrix inner product. The formulation involves computing the outer product between the prediction error vector (e.g., $\mathbf{e}_{y_{i,k'}} - \pi_\theta(\cdot|\boldsymbol{x}, \boldsymbol{y}_{i,<k'})$) and the hidden embedding, which incurs a computational complexity of $O(|\mathcal{V}|d)$. Since the probability mass is primarily concentrated on samples' words, we restrict the computation to the vocabulary $\mathcal{V}_{\mathcal{D}}$ associated with samples' words. Given that $|\mathcal{V}_{\mathcal{D}}| \ll |\mathcal{V}|$, this significantly reduces the overall cost to $O(|\mathcal{V}_{\mathcal{D}}|d)$ (see detailed efficiency comparison in Tab. 6). Notably, when performing per-batch valuation calculations, the vocabulary size can be further decreased to the in-batch vocabulary size, as demonstrated in step 6 of Algorithm 1.

---

[1]This assumption is mild, as training inputs often differ from valuation inputs in practice. E.g., in vision language tasks, images are often unique or paired with different questions. More discussion see Appendix.

**For-Value Algorithm:** Algorithm 1 summarizes our efficient batch computation of For-Value. We first extract hidden embeddings and prediction errors via a single forward pass over the valuation and training batches. Restricting calculations to the in-batch vocabulary and batching the computations significantly reduces overhead while preserving accuracy. Finally, we sort the scores to rank the training samples according to their estimated influence. Importantly, the algorithm can be naturally extended to a group of valuation pairs by averaging their influence scores. The complete pipeline is depicted in Fig. 2.

## 5 EXPERIMENT SETUP

In this section, we describe the experimental setup. More details please see Appendix.

**Baseline Methods.** We focus on the comparison with baseline methods designed for efficiency: Hessian-free (Pruthi et al., 2020; Charpiat et al., 2019) estimates influence scores via the dot product of first-order gradients, which is equivalent to the Trace-Inf (Pruthi et al., 2020) or the first-order in-run Shapley (Wang et al., 2024a) at the last training iteration. DataInf (Kwon et al., 2024) uses a Hessian approximation tailored for parameter-efficient fine-tuning, while HyperINF (Zhou et al., 2024) employs a low-rank Fisher approximation of the Hessian. Finally, we include an embedding similarity method (Yang et al., 2025), originally proposed for image generation models, denoted as Emb.

---

**Algorithm 1** For-Value: Forward-Only Data Valuation

**Input:** Training set $\{(\boldsymbol{x}_i, \boldsymbol{y}_i)\}_{i=1}^N$; valuation pair $(\boldsymbol{x}_v, \boldsymbol{y}_v)$; model $\pi_\theta$; batch size $B$.
**Output:** Data valuation $\mathcal{S}$.

1: Compute $\{\boldsymbol{h}_{\boldsymbol{x}_v, \boldsymbol{y}_{v,<k}}\}_{k=1}^{|\boldsymbol{y}_v|}$ and $\{\pi_\theta(\cdot|\boldsymbol{x}_v, \boldsymbol{y}_{v,<k})\}_{k=1}^{|\boldsymbol{y}_v|}$ by doing inference $\pi_\theta(\boldsymbol{x}_v, \boldsymbol{y}_v)$.
2: **for** each batch $\{(\boldsymbol{x}_j, \boldsymbol{y}_j)\}_{j=1}^B$ **do**
3:    Compute $\{\boldsymbol{h}_{\boldsymbol{x}_j, \boldsymbol{y}_{j,<k'}}\}_{k'=1}^{|\boldsymbol{y}_j|}$ and
4:    $\{\pi_\theta(\cdot|\boldsymbol{x}_j, \boldsymbol{y}_{j,<k'})\}_{k'=1}^{|\boldsymbol{y}_j|}$ by running batch inference.
5:    $\hat{\mathcal{V}} \leftarrow \bigcup_{j=1}^B \mathcal{V}_{\boldsymbol{x}_j, \boldsymbol{y}_j} \cup \mathcal{V}_{\boldsymbol{x}_v, \boldsymbol{y}_v}$
6:    Compute errors $(\mathbf{e} - \pi(\cdot))$ for tokens in $\hat{\mathcal{V}}$.
7:    For each in batch, compute $S_{v,j}$ via Eq. (2).
8: **end for**
9: $\mathcal{S} \leftarrow \{(\boldsymbol{x}_i, \boldsymbol{y}_i, S_{v,i})\}_{i=1}^N$.
10: Sort $\mathcal{S}$ by $S_{v,i}$ (descending).
11: **return** $\mathcal{S}$.

---

**Models.** Following Kwon et al. (2024), we evaluate LLMs using Llama-2-13B-chat (Touvron et al., 2023) and Qwen-2.5-1.5B (Qwen et al., 2025) to cover a wider range of model sizes and families. Moreover, thanks to the efficiency of our method, we are able to run For-Value on Qwen2.5 series models from 7B up to 72B parameters. In contrast, baseline methods require extensive training and prolonged runtimes, making them costly for these larger models. For VLMs, we adopt the widely used Qwen-2.5-VL-3B-Instruct (Bai et al., 2025a) and Llama-3.2-11B-Vision (Meta, 2024).

**Influential Data Identification Tasks.** We evaluate all methods on influential data identification for LLMs and VLMs, following Kwon et al. (2024). For LLMs, we use sentence transformation and math word problem datasets (with and without reasoning). For VLMs, we adapt image-to-text tasks from Kwon et al. (2024) to an image-to-text generation setting, including style generation (cartoons, pixel art, line sketches) and subject generation using the DreamBooth dataset (Ruiz et al., 2023). We adopt two evaluation metrics from Kwon et al. (2024): (i) AUC, measuring the correlation between data values and pseudo-labels (1 if training and valuation samples share a class, 0 otherwise), averaged over valuation points; and (ii) Recall, the proportion of top-ranked training samples sharing the same class as the valuation point. More details and dataset examples see Appendix Sec. A.7.

**Mislabeled Data Detection Tasks.** We evaluate mislabeled data detection on VLMs using the Kaggle cat–dog dataset (kag, 2013), reformulated as a QA task with 50% label being flipped, and report AUC and Recall; examples and further details are provided in the Appendix Sec. A.7.

**Data Selection For Finetuning.** We evaluate the practical utility of For-Value across two key reasoning domains: mathematics and medicine. For mathematics, we use the GSM8K (Cobbe et al., 2021) dataset to assess influential data identification, while for medicine, we employ the Noise-Huatuo-Complex-CoT (Chen et al., 2024) dataset to examine robustness under noisy training. We further extend our study to vision–language models by applying For-Value to PMC-Reasoning Huang et al. (2025). More details for each task are provided in Appendix Sec. A.5.

**Efficiency Evaluation.** For influential and mislabeled data detection with models under 32B, we compute data values using a single A100 (80G) GPU with identical hardware settings. For fine-tuning data selection, we use a single H100 (96G) GPU to calculate the data value for fair comparison. More details please see Appendix Sec. A.3.

| Method | Qwen2.5-1.5B | | Llama-2-13B-chat | |
|---|---|---|---|---|
| | AUC ↑ | Recall ↑ | AUC ↑ | Recall ↑ |
| **Sentence transformations** | | | | |
| Hessian-free(Pruthi et al., 2020) | $0.785 \pm 0.096$ | $0.370 \pm 0.139$ | $0.999 \pm 0.002$ | $0.985 \pm 0.033$ |
| DataInf(Kwon et al., 2024) | $0.981 \pm 0.019$ | $0.826 \pm 0.121$ | $\mathbf{1.000 \pm 0.000}$ | $\underline{0.997 \pm 0.010}$ |
| HyperINF(Zhou et al., 2024) | $\underline{0.993 \pm 0.013}$ | $\underline{0.934 \pm 0.063}$ | $\mathbf{1.000 \pm 0.000}$ | $\underline{0.998 \pm 0.011}$ |
| Emb(Yang et al., 2025) | $0.546 \pm 0.306$ | $0.148 \pm 0.205$ | $0.854 \pm 0.192$ | $0.563 \pm 0.412$ |
| For-Value (ours) | $\mathbf{1.000 \pm 0.001}$ | $\mathbf{0.989 \pm 0.025}$ | $\mathbf{1.000 \pm 0.000}$ | $\mathbf{1.000 \pm 0.001}$ |
| **Math Problem (w/o reasoning)** | | | | |
| Hessian-free(Pruthi et al., 2020) | $0.835 \pm 0.235$ | $0.592 \pm 0.291$ | $0.770 \pm 0.174$ | $0.258 \pm 0.388$ |
| DataInf(Kwon et al., 2024) | $0.985 \pm 0.032$ | $0.878 \pm 0.154$ | $\mathbf{1.000 \pm 0.000}$ | $\underline{0.999 \pm 0.006}$ |
| HyperINF(Zhou et al., 2024) | $\underline{0.986 \pm 0.024}$ | $\underline{0.942 \pm 0.080}$ | $0.995 \pm 0.018$ | $\underline{0.967 \pm 0.057}$ |
| Emb(Yang et al., 2025) | $0.555 \pm 0.298$ | $0.146 \pm 0.295$ | $0.762 \pm 0.239$ | $0.389 \pm 0.477$ |
| For-Value (ours) | $\mathbf{1.000 \pm 0.000}$ | $\mathbf{0.998 \pm 0.011}$ | $\mathbf{1.000 \pm 0.000}$ | $\mathbf{1.000 \pm 0.002}$ [2] |
| **Math Problem (w/ reasoning)** | | | | |
| Hessian-free(Pruthi et al., 2020) | $0.829 \pm 0.172$ | $0.524 \pm 0.350$ | $0.772 \pm 0.173$ | $0.258 \pm 0.388$ |
| DataInf(Kwon et al., 2024) | $0.987 \pm 0.030$ | $0.892 \pm 0.155$ | $\underline{1.000 \pm 0.001}$ | $\underline{0.996 \pm 0.025}$ |
| HyperINF(Zhou et al., 2024) | $\underline{0.988 \pm 0.023}$ | $\underline{0.950 \pm 0.060}$ | $0.994 \pm 0.018$ | $0.961 \pm 0.074$ |
| Emb(Yang et al., 2025) | $0.560 \pm 0.310$ | $0.198 \pm 0.311$ | $0.725 \pm 0.217$ | $0.270 \pm 0.420$ |
| For-Value (ours) | $\mathbf{1.000 \pm 0.000}$ | $\mathbf{0.998 \pm 0.008}$ | $\mathbf{1.000 \pm 0.000}$ | $\mathbf{1.000 \pm 0.000}$ |

Table 1: Influential data identification results on LLMs. For-Value consistently achieves comparable or superior performance. Results are reported as Mean ± Standard Deviation (std).

# 6 RESULTS

In this section, we detail the results of For-Value and baselines on LLMs and VLMs.

## 6.1 INFLUENTIAL & MISLABELED DATA IDENTIFICATION

**Influential data identification Results on LLM.** We first present the results for text generation tasks in Tab. 1, where For-Value consistently matches or outperforms all baseline methods across the evaluated LLM benchmarks:

(1) *Sentence Transformation:* As shown in Tab. 1, for the sentence transformation task, For-Value achieves perfect or near-perfect AUC and recall scores for both models. Notably, on Qwen2.5-1.5B, For-Value surpasses the strongest baseline, HyperINF, by 0.7% in AUC and by 6.5% in recall.

(2) *Math Problems (w/&w/o reasoning):* A similar pattern holds for the math problem tasks, both with and without reasoning (data samples in Tab. 9). As shown in Tab. 1, For-Value delivers higher-quality influence identification

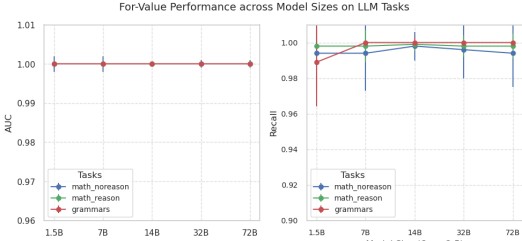

Figure 3: For-Value performance across model sizes and tasks (Mean±std).

with just a single forward pass, improving recall by about 6% over the best-performing baseline HyperINF on both math datasets with Qwen model.

These results demonstrate that For-Value reliably identifies influential data points across different tasks and model scales, combining strong accuracy with practical efficiency.

**Influential data identification Results on VLM.** We next report the results on VLMs in Tab. 2. (1) For subject generation, For-Value achieves the highest AUC and recall scores for both Qwen-2.5-VL-3B-Instruct and Llama-3.2-11B-Vision, consistently outperforming all baselines. Specifically, For-Value exceeds the strongest baseline, HyperINF, by more than 7% in recall for both models for the 11B model. (2) In the more challenging style generation task, For-Value demonstrates a clear advantage, with AUC improvements of over 0.35 compared to the baselines, and even larger gains over the Emb method. Notably, the performance of baselines drops more sharply on this task, raising concerns on their robustness on complex dataset. These findings confirm that For-Value

---

[2]AUC and Recall values reported as 1.0 may still include a non-zero std due to rounding. The large std arise because the underlying value distribution is highly polarized, clustering near either 1 or 0.

effectively identifies influential data points for VLMs across diverse tasks and model sizes.

| Method | Qwen2.5-VL-3B-Instruct | | Llama-3.2-11B-vision | |
|---|---|---|---|---|
| | AUC ↑ | Recall ↑ | AUC ↑ | Recall ↑ |
| **Image-to-text subject generation** | | | | |
| Hessian-free(Pruthi et al., 2020) | $0.979 \pm 0.038$ | $0.738 \pm 0.399$ | $0.961 \pm 0.093$ | $0.765 \pm 0.365$ |
| DataInf(Kwon et al., 2024) | $0.989 \pm 0.024$ | $0.836 \pm 0.318$ | $0.958 \pm 0.119$ | $0.797 \pm 0.323$ |
| HyperINF(Zhou et al., 2024) | $0.988 \pm 0.047$ | $\mathbf{0.902 \pm 0.220}$ | $0.993 \pm 0.025$ | $0.919 \pm 0.186$ |
| Emb(Yang et al., 2025) | $0.841 \pm 0.189$ | $0.206 \pm 0.458$ | $0.841 \pm 0.189$ | $0.206 \pm 0.379$ |
| For-Value (ours) | $\mathbf{0.994 \pm 0.018}$ | $0.897 \pm 0.287$ | $\mathbf{0.995 \pm 0.040}$ | $\mathbf{0.985 \pm 0.068}$ |
| **Image-to-text style generation** | | | | |
| Hessian-free(Pruthi et al., 2020) | $0.515 \pm 0.096$ | $0.799 \pm 0.162$ | $0.515 \pm 0.079$ | $0.824 \pm 0.145$ |
| DataInf(Kwon et al., 2024) | $0.520 \pm 0.094$ | $0.760 \pm 0.181$ | $0.515 \pm 0.174$ | $0.785 \pm 0.164$ |
| HyperINF(Zhou et al., 2024) | $0.516 \pm 0.055$ | $0.860 \pm 0.103$ | $0.490 \pm 0.090$ | $0.821 \pm 0.137$ |
| Emb(Yang et al., 2025) | $0.560 \pm 0.310$ | $0.198 \pm 0.311$ | $0.553 \pm 0.294$ | $0.340 \pm 0.467$ |
| For-Value (ours) | $\mathbf{0.895 \pm 0.138}$ | $\mathbf{0.916 \pm 0.153}$ | $\mathbf{0.974 \pm 0.059}$ | $\mathbf{0.997 \pm 0.013}$ |
| **Mislabeled Data Detection** | | | | |
| Hessian-free(Pruthi et al., 2020) | $0.719 \pm 0.098$ | $0.760 \pm 0.088$ | $0.962 \pm 0.019$ | $0.955 \pm 0.068$ |
| DataInf(Kwon et al., 2024) | $0.760 \pm 0.088$ | $0.901 \pm 0.147$ | $\mathbf{1.000 \pm 0.000}$ | $1.000 \pm 0.003$ |
| HyperINF(Zhou et al., 2024) | $0.770 \pm 0.077$ | $0.916 \pm 0.128$ | $1.000 \pm 0.001$ | $1.000 \pm 0.006$ |
| Emb(Yang et al., 2025) | $0.741 \pm 0.061$ | $0.533 \pm 0.075$ | $0.933 \pm 0.044$ | $0.996 \pm 0.015$ |
| For-Value (ours) | $\mathbf{0.885 \pm 0.055}$ | $\mathbf{0.999 \pm 0.010}$ | $0.995 \pm 0.008$ | $\mathbf{1.000 \pm 0.000}$ |

Table 2: Influential data identification and mislabeled data detection performance for different VLM tasks. For-Value consistently delivers comparable or superior performance in identifying influential data and detecting mislabeled data across various VLM tasks compared to baseline methods.

**Mislabeled Data Detection.** Our mislabeled data detection results in Tab. 2 demonstrate For-Value's strong performance across model scales. On the Qwen-VL-3B model, For-Value achieves an 11.5% higher AUC and an 8.3% higher Recall compared to the best baseline (HyperINF), showing significant improvements in identifying mislabeled examples. The method performs equally well on the larger Llama-3.2-11B model, matching the near-perfect detection rates (AUC > 0.99, Recall = 1.0) of gradient-based approaches. This consistent performance across both small (3B) and large (11B) VLMs highlights For-Value's scalability and effectiveness. Notably, For-Value achieves these results using just a single forward pass, requiring seconds rather than the hours needed by baseline methods.

## 6.2 DATA SELECTION FOR FINETUNING

Having established the strong performance of For-Value in identifying both influential and noisy data, we next assess its practical utility on mathematics and medicine. Given poor performance of Emb in prior experiments, we excluded it from these evaluations.

**Mathematics: GSM8K.** We begin by examining influential data identification on the GSM8K (Cobbe et al., 2021) dataset, which provides ground-truth training and test reasoning pairs. This setup enables us to select high-value training samples and measure their effect on test accuracy. Following (Deng et al., 2024b), we report greedy performance in Tab. 3, where

| Llama-3.1-8B | GSM8K (1%) ↑ | GSM8K (5%) ↑ | Time ↓ |
|---|---|---|---|
| Full (100%) | | 47.8 | – |
| Hessian-free | 41.5 | 41.8 | 1.4 h |
| HyperINF | 41.9 | 42.8 | 2.4 h |
| DataInf | 41.7 | 42.0 | 1.9 h |
| For-Value (ours) | **45.2** | **48.3** | **0.3 h** |

Table 3: GSM8K greedy decoding accuracy of Llama-3.1-8B. Best results are in **bold**.

fine-tuning on the top 5% most influential samples selected by For-Value achieves the highest accuracy of 48.3%, surpassing the strongest baseline, HyperINF, by 5.5% and even slightly outperforming training on the full dataset. When the selection rate is further reduced to 1%, performance decreases as expected, but For-Value still exceeds all baselines by up to 3.3%. Crucially, For-Value also provides the most efficient valuation, requiring only 0.3 hours, more than 5× faster than baselines.

**Medicine: Noise-Huatuo-Complex-CoT.** To examine robustness under noisy training conditions, we construct a corrupted version of the Huatuo-Complex-CoT dataset (Chen et al., 2024). We ran-

domly sample 5,000 examples without replacement and inject noise into 40% of them by inserting or removing irrelevant words (examples see Fig. 7 in Appendix), resulting in the Noise-Huatuo-Complex-CoT dataset. Another 5,000 clean examples are reserved for valuation, and models are evaluated on five held-out medical QA test sets. Within this setting, we apply `For-Value` and competing methods to select high-quality training subsets for fine-tuning. As shown in Tab. 4, `For-Value` consistently delivers the strongest results. With only 5% data, it reaches an average accuracy of 60.31%, outperforming the best baseline (`DataInf`) by 3%. At 10%, `For-Value` shows an even clearer advantage, achieving the best score across all tasks with an average of 62.35%, exceeding the strongest baseline `HyperINF` by 2.1%. Crucially, `For-Value` also provides the most efficient valuation, requiring only 0.8h, up to 6× faster than baselines. These results underscore the effectiveness of `For-Value` in identifying valuable data even when training data is noisy. More analysis are provided in Appendix Sec. A.6.

| Method (Llama-3.1-8B-Ins) | MedQA | MedMCQA | PubMedQA | MMLU-Pro-med | GPQA-med | Average ↑ | Time ↓ |
|---|---|---|---|---|---|---|---|
| Base | 56.84 | 61.90 | 77.00 | 59.02 | 44.35 | 59.82 | – |
| *5% Data* | | | | | | | |
| Hessian-free | 55.41 | 58.05 | 73.40 | 54.53 | 38.46 | 55.97 | 2.0 (h) |
| HyperINF | 55.15 | 57.58 | 71.50 | 54.14 | 43.08 | 56.29 | 5.3 (h) |
| DataInf | 55.39 | 57.74 | 73.30 | 54.07 | 45.13 | 57.13 | 4.1 (h) |
| For-Value (ours) | **56.80** | **62.92** | **77.60** | **58.31** | **45.90** | **60.31** | **0.8 (h)** |
| *10% Data* | | | | | | | |
| Hessian-free | 57.02 | 59.15 | 72.30 | 57.13 | 47.69 | 58.66 | 2.0 (h) |
| HyperINF | 56.94 | 62.76 | 77.40 | 57.85 | 48.46 | 60.28 | 5.3 (h) |
| DataInf | 56.61 | 61.74 | 75.60 | 56.81 | 43.85 | 58.92 | 4.1 (h) |
| For-Value (ours) | **57.61** | **67.16** | **78.30** | **58.18** | **50.51** | **62.35** | **0.8 (h)** |

Table 4: Results of data selection for fine-tuning on the Noise Huatuo-Complex-CoT (Llama-3.1-8B-Ins).

**Medical VQA.** To evaluate the effectiveness of `For-Value` on vision–language models, we conduct experiments on the PMC-Reasoning dataset (Huang et al., 2025; Zhang et al., 2023). We randomly sample 10,000 examples for training and 5,000 for valuation without replacement. Fine-tuning subsets are then selected from the training pool using `For-Value` as well as baseline methods, and the resulting models are fine-tuned and evaluated on six held-out test sets. As shown in Tab. 5, `For-Value` delivers the strongest overall performance. With 10% data, it achieves the highest average accuracy (52.23%), exceeding the base model by over 3% and the best-performing baseline, `HyperINF`, by 0.6%. At 20% data, `For-Value` maintains competitive performance (52.67%), ranking second only to `HyperINF`. Importantly, `For-Value` consistently achieves these results with the lowest computational cost (0.4h vs. 1.6–1.7h for baseline methods). Notably, all data valuation methods surpass full fine-tuning, highlighting the benefit of selecting high-value subsets for training. Overall, these results demonstrate that `For-Value` reliably identifies influential data for medical VQA while offering significant efficiency gains.

| Method (Qwen2.5-VL-3B) | MMMU | MedX-M | PathVQA | PMC | SLAKE | VQA-Rad | Average ↑ | Time ↓ |
|---|---|---|---|---|---|---|---|---|
| Base | 44.12 | 20.69 | 61.96 | 44.77 | 61.30 | 62.01 | 49.14 | – |
| Full (Huang et al., 2025) | 47.84 | 21.46 | 52.76 | 54.55 | 65.79 | 58.58 | 50.16 | – |
| *10% Data* | | | | | | | | |
| Hessian-free | 48.82 | 20.65 | 61.18 | 49.60 | 61.78 | 63.60 | 50.94 | 1.3 (h) |
| HyperINF | **50.00** | 21.60 | 61.10 | 50.45 | 62.50 | 63.97 | 51.60 | 1.7 (h) |
| DataInf | 49.41 | 21.10 | 62.64 | **50.55** | 59.38 | **65.81** | 51.48 | 1.6 (h) |
| For-Value (ours) | 47.06 | **23.05** | **62.93** | 49.55 | **67.55** | 63.24 | **52.23** | **0.4 (h)** |
| *20% Data* | | | | | | | | |
| Hessian-free | 52.94 | 21.40 | 61.81 | 52.05 | 63.46 | 62.50 | 52.36 | 1.3 (h) |
| HyperINF | **56.47** | 20.50 | **62.14** | **51.45** | 62.98 | **64.71** | **53.04** | 1.7 (h) |
| DataInf | 48.82 | 21.25 | 62.58 | 51.35 | 63.46 | 63.24 | 51.78 | 1.6 (h) |
| For-Value (ours) | 54.12 | **22.45** | 60.26 | 50.45 | **65.14** | 63.60 | 52.67 | **0.4 (h)** |

Table 5: Results of data selection for fine-tuning on the PMC-Reasoning dataset. Best results are in **bold**, and second-best are underlined.

### 6.3 ABLATION STUDY & EFFICIENCY

In this section, we present ablation and efficiency studies based on influential and mislabeled data identification tasks in Sec. 6.1.

**Effect of prediction error similarity $\alpha$.** We perform an ablation study to evaluate the role of the $\alpha$ term by setting $\alpha$ to 1 in the computation of Eq. (2). This simplification reduces the score to $\left\langle \sum_{k=1}^{|\boldsymbol{y}_v|} \boldsymbol{h}_{\boldsymbol{x}_v, \boldsymbol{y}_{v,<k}}, \sum_{k'=1}^{|\boldsymbol{y}_i|} \boldsymbol{h}_{\boldsymbol{x}_i, \boldsymbol{y}_{i,<k'}} \right\rangle$, which measures contextualized text embedding similarity between two data samples' $\boldsymbol{y}$ (context is the input $\boldsymbol{x}$ and notably, in practice, in text generation it is the whole text and text part for image-to-text generation.). This is equivalent to the `Emb` baseline. As shown in Tab. 1 and Tab. 2, `For-Value` consistently and significantly outperforms `Emb` across both LLM and VLM tasks. This highlights the importance of including $\alpha$ in the calculation. Intuitively, the prediction error in $\alpha$ term acts as a token-level weight: when the model's confidence for a token in the training data is already high, its prediction error is small and contributes little gradient signal (loss is small); similarly, when the valuation token is predicted with high confidence, any further increase in its probability is limited, implying that it is less influenced by the training data. While `Emb` performs well for data valuation in generative image models, its degraded performance shows that directly applying it to LLMs/VLMs is ineffective due to a different training objective.

**`For-Value` Performance across model sizes.** Fig. 3 shows that `For-Value` maintains consistently high performance across different model sizes and tasks. Both AUC and Recall stay close to 1.0 for all tasks, indicating that scaling up the model does not degrade effectiveness. This stability confirms that `For-Value` generalizes well to larger models while preserving accuracy, making it reliable for practical deployment on a range of LLM tasks.

**Time Cost Analysis.** To further demonstrate efficiency, we compare the time cost of `For-Value` with that of the baselines across different model sizes and tasks. As shown in Fig. 4a, `For-Value` maintains consistently low runtime, even as model size increases from 1.5B to 72B parameters. For all tasks, the runtime remains within a few hundred seconds, highlighting its practical scalability. In contrast, as shown in Fig. 4b, baseline methods for the sentence transformation task require significantly more time—measured in hours rather than seconds. The best-performing baseline, `HyperINF`, becomes especially

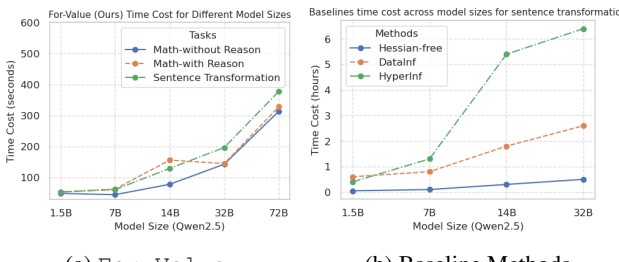

(a) `For-Value`.      (b) Baseline Methods

Figure 4: Time cost analysis: (a) Time cost of `For-Value` across different model sizes and tasks. (b) Time cost of baseline methods on sentence transformation task across different model sizes. Notably, `For-Value` is significantly more efficient than the baselines, with time costs measured in seconds, whereas the baselines require up to several hours.

costly for larger models, taking up to 6 hours for the 32B model. This underscores the efficiency advantage of `For-Value`, which delivers competitive or superior performance with minimal computational cost. More discuss on efficiency please see Appendix Sec. A.4.

## 7 CONCLUSION

In this work, we presented `For-Value`, a forward-only data valuation framework specifically designed for pretrained LLMs and VLMs. By relying solely on a single forward pass to estimate per-sample influence, `For-Value` removes the computational bottlenecks associated with gradient and Hessian calculations. Our theoretical analysis grounds `For-Value` in the learning dynamics of autoregressive modeling, providing a solid foundation for its effectiveness. Extensive experiments across tasks and model scales show that `For-Value` matches or surpasses state-of-the-art baselines in identifying mislabeled and influential samples. It also selects higher-value subsets for fine-tuning, yielding a better performance in mathematical and medical domains. Crucially, these gains are achieved with substantial improvements in computational efficiency, highlighting `For-Value` as a practical and scalable solution for data valuation in large foundation models.

## 8 ETHICS STATEMENT

This work complies with the ICLR Code of Ethics. All datasets used are publicly available and utilized under their respective licenses, with no involvement of human subjects or sensitive data. License details are provided in Sec. A.10.

## 9 REPRODUCIBILITY STATEMENT

We ensure reproducibility by providing full method details in Sec. 4, with proofs in Sec. A.2. Experimental settings and datasets are described in Sec. 5 and Appendix.

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

CONTENTS

# A    APPENDIX

## A.1    TRAINING LOSS OF LLMS AND VLMS

To adapt a pretrained LLM or VLM to a specific domain or task, models are typically trained on a supervised dataset $\mathcal{D} = (\boldsymbol{x}_i, \boldsymbol{y}_i)_{i=1}^n$ of input-output pairs. Training is commonly performed using the standard teacher-forcing objective, which minimizes the negative log-likelihood of the target sequence:

$$\mathcal{L}_{\mathrm{SFT}}(\theta) = -\frac{1}{n}\sum_{i=1}^n \ln \pi_\theta(\boldsymbol{y}_i|\boldsymbol{x}_i) - \frac{1}{n}\sum_{i=1}^n \sum_{k=1}^{|\boldsymbol{y}_i|} \ln \pi_\theta(y_{i,k}|\boldsymbol{x}_i, \boldsymbol{y}_{i,<k}).$$

This objective maximizes the likelihood that the model generates the correct output sequence conditioned on the input and the ground-truth prefix at each step. The parameters are updated using gradient descent or its variants:

$$\theta \leftarrow \theta - \eta\,\nabla_\theta\,\mathcal{L}_{\mathrm{SFT}}(\theta), \quad \text{with} \quad \theta_{t=0} = \theta_0,$$

where $\eta > 0$ is the learning rate. Teacher forcing stabilizes fine-tuning by supplying the true prefix $\boldsymbol{y}_{<k}$ during training, enabling the model to align its predictions closely with the target data distribution in the new domain.

## A.2    PROOF OF THEOREM 1

In this section, we give the detailed proof of our Theorem 1, we start by proving the following theorem:

**Theorem 2.** *For a data $\boldsymbol{x}_v$ and its generation $\boldsymbol{y}_v$ that await valuation, at any time $t \geq 0$ of training using a training data $(\boldsymbol{x}_i, \boldsymbol{y}_i), i \in [n]$, the training data exhibits larger value to the valuation data as the following increases:*

$$\sum_{k=1}^{|\boldsymbol{y}_v|}\sum_{k'=1}^{|\boldsymbol{y}_i|} \alpha_{k,k'}(t) \cdot \left\langle \mathbf{h}_{\boldsymbol{x}_v, \boldsymbol{y}_{v,<k}}(t), \mathbf{h}_{\boldsymbol{x}_i, \boldsymbol{y}_{i,<k'}}(t) \right\rangle +$$

$$\sum_{k=1}^{|\boldsymbol{y}_v|} \left\langle \boldsymbol{w}_{\boldsymbol{y}_{v,k}}(t) - \sum_{z \in \mathcal{V}} \pi_{\theta(t)}(z|\boldsymbol{x}_v) \cdot \boldsymbol{w}_z(t), \left(\boldsymbol{w}_{\boldsymbol{y}_{i,k}} - \sum_{z \in \mathcal{V}} \pi_{\theta(t)}(z|\boldsymbol{x}_v) \cdot \boldsymbol{w}_z(t)\right) \right\rangle \qquad (3)$$

*Proof.*

$$\frac{d}{dt} \ln \pi_{\theta(t)}(\mathbf{y}_v|\mathbf{x}_v) = \left\langle \nabla \ln \pi_{\theta(t)}(\mathbf{y}_v|\mathbf{x}_v), \frac{d}{dt}\theta(t) \right\rangle$$

$$= \left\langle \nabla \ln \pi_{\theta(t)}(\mathbf{y}_v|\mathbf{x}_v), -\eta\nabla\mathcal{L}_D(\theta) \right\rangle$$

$$= \left\langle \nabla \ln \pi_{\theta(t)}(\mathbf{y}_v|\mathbf{x}_v), \eta\sum_{i=1}^n \nabla \ln \pi_{\theta(t)}(y_i|\mathbf{x}_i) \right\rangle$$

As per the unconstrained features Assumption, the model's trainable parameters are

$$\theta = \left(\boldsymbol{W}, \mathbf{h}_{\boldsymbol{x}_v}, \left\{\mathbf{h}_{\boldsymbol{x}_v, \boldsymbol{y}_{v,<k}}\right\}_{k \in \{2,...,|\boldsymbol{y}_v|\}}, \left\{\mathbf{h}_{\boldsymbol{x}_i, \boldsymbol{y}_{i,<k'}}\right\}_{i \in [n], k' \in \{1,...,|\boldsymbol{y}_i|\}}\right).$$

Unfolding the gradients with respect to these parameters yields:

$$\frac{d}{dt} \ln \pi_{\theta(t)}(\boldsymbol{y}_v|\boldsymbol{x}_v) = \left\langle \nabla_{\boldsymbol{W}} \ln \pi_{\theta(t)}(\boldsymbol{y}_v|\boldsymbol{x}_v), \sum_i^n \nabla_{\boldsymbol{W}} \ln \pi_{\theta(t)}(\boldsymbol{y}_i|\boldsymbol{x}_i) \right\rangle$$

$$+ \underbrace{\sum_{k=1}^{|\boldsymbol{y}_v|} \left\langle \nabla_{\mathbf{h}_{\boldsymbol{x}_v, \boldsymbol{y}_{v,<k}}} \ln \pi_{\theta(t)}(\boldsymbol{y}_{v,k}|\boldsymbol{x}_v, \boldsymbol{y}_{v,<k}), \sum_{i'=1}^{n_k} \nabla_{\mathbf{h}_{\boldsymbol{x}_v, \boldsymbol{y}_{v,<k}}} \ln \pi_{\theta(t)}(\boldsymbol{y}_{i',k}|\boldsymbol{y}_{v,<k}) \right\rangle}_{\text{(II) Training data have the same } (\boldsymbol{x}_v, \boldsymbol{y}_{v,<k})} .$$

$$(4)$$

where $n_k$ is the number of training data whose input and prediction before token $k$ are the same as valuation data $(\boldsymbol{x}_v, \boldsymbol{y}_{v,<k})$. Since we have

$$\nabla_{\mathbf{W}} \ln \pi_{\theta(t)}(z|\mathbf{x}) = \left( \mathbf{e}_z - \sum_{z' \in \mathcal{V}} \pi_{\theta(t)}(z'|\mathbf{x}) \cdot \mathbf{e}_{z'} \right) \mathbf{h}_{\mathbf{x}}^\top(t),$$

$$\nabla_{\mathbf{h}_{\mathbf{x}}} \ln \pi_{\theta(t)}(z|\mathbf{x}) = \mathbf{W}_z(t) - \sum_{z' \in \mathcal{V}} \pi_{\theta(t)}(z'|\mathbf{x}) \cdot \mathbf{W}_{z'}(t).$$

Putting this back in (4) together with a few algebra steps, yields

$$\frac{d}{dt} \ln \pi_{\theta(t)}(\boldsymbol{y}_v|\boldsymbol{x}_v) = \text{(I)} + \text{(II)} \tag{5}$$

where:

$$\text{(I)} = \sum_{k=1}^{|\boldsymbol{y}_v|} \sum_{i=1}^n \sum_{k'=1}^{|\boldsymbol{y}_i|} \alpha_{k,k'}(t) \cdot \left\langle \mathbf{h}_{\boldsymbol{x}_v, \boldsymbol{y}_{v,<k}}(t), \mathbf{h}_{\boldsymbol{x}_i, \boldsymbol{y}_{i,<k'}}(t) \right\rangle \tag{6}$$

$$\text{(II)} = \sum_{k=1}^{|\boldsymbol{y}_v|} \left\langle \boldsymbol{w}_{\boldsymbol{y}_{v,k}}(t) - \sum_{z \in \mathcal{V}} \pi_{\theta(t)}(z|\boldsymbol{x}_v) \cdot \boldsymbol{w}_z(t), \sum_{i'=1}^{n_k} (\boldsymbol{w}_{\boldsymbol{y}_{i',k}} - \sum_{z \in \mathcal{V}} \pi_{\theta(t)}(z|\boldsymbol{x}_v) \cdot \boldsymbol{w}_z(t)) \right\rangle \tag{7}$$

where $\alpha_{k,k'}(t) = \left\langle \mathbf{e}_{\boldsymbol{y}_{v,k}} - \pi_{\theta(t)}(\cdot|\boldsymbol{x}, \boldsymbol{y}_{v,<k}), \mathbf{e}_{\boldsymbol{y}_{i,k'}} - \pi_{\theta(t)}(\cdot|\boldsymbol{x}, \boldsymbol{y}_{i,<k'}) \right\rangle$. By taking the $i$-th sample, we can obtain Theorem 2. $\qquad\square$

We observe the following:

(1) When the training input $\boldsymbol{x}_i$ differs from the valuation input $\boldsymbol{x}_v$, its influence on the valuation target arises solely through Term (I), which captures the contribution of the token embeddings and all network parameters except the token unembedding layer.

(2) The effect of the token unembeddings is concentrated in cases where the training and valuation data share the same input $\boldsymbol{x}$ and exhibit overlapping output predictions $\boldsymbol{y}$.
To eliminate this dependence on token unembeddings, we impose the following assumption:

**Assumption 2** (Distinct Input). *The training dataset satisfies that no training input $\boldsymbol{x}_i$ is identical to the valuation input $\boldsymbol{x}_v$.*

Under the Assumption 2, the contribution from token unembeddings (Term (II)) vanishes, so that the influence of the training data on the valuation data arises entirely through the shared representation features captured in Term (I). This assumption is mild, as training inputs typically differ from valuation inputs in practice — especially in vision-language datasets, where the input images are almost always distinct. Extending this result to cases where training examples share the same input but differ in their outputs $\boldsymbol{y}$ is straightforward: the output prefix $\boldsymbol{y}_{<k}$ can be incorporated into the input $\boldsymbol{x}$, treating each unique pair $(\boldsymbol{x}, \boldsymbol{y}_{<k})$ as a distinct input, where $k-1$ indicates the point at which the outputs begin to differ. Combining Theorem 2 and Assumption 2 then yields Theorem 1.

### A.3 ADDITIONAL DETAILS OF INFLUENTIAL AND MISLABELED DATA DETECTION

**Training setting for baselines.** While `For-Value` requires only a single forward pass, the influence function-based baselines `Hessian-free` and `DataInf` require fine-tuning the models to

convergence. For text generation tasks, we follow the training setup in Kwon et al. (2024), except to llama-2-13B, we use float16 weights instead of 8-bit quantization. For image-to-text generation tasks, we apply LoRA to every query and value matrix within the model's attention layers. To fine-tune VLMs, we use a learning rate of $2 \times 10^{-4}$, LoRA hyperparameters $r = 8$ and $\alpha = 32$, float16 model weights, a batch size of 32, and train for 20 epochs.

**Efficiency details.** For larger 32B and 72B models in Fig. 4, we employ 4 A100 GPUs for inference and a single A100 for value computation. Baseline methods requiring training are fine-tuned on up to 8 GPUs, with the 32B model quantized to 8-bit to enable valuation on a single A100. Due to their long runtime, we restrict baselines to the sentence transformation task and, for 14B/32B models, sample 10% of valuation data—scaling time by a factor of 10 to estimate totals. Despite these adjustments, `For-Value` achieves substantially lower runtime without quantization and with fewer GPUs.

## A.4 ADDITIONAL RESULTS

**Complexity Analysis.** Tab. 6 compares the training, computational, and memory costs of different methods. Traditional approaches such as IF, `Hessian-free`, `HyperINF`, and `DataInf` rely on gradient traces or Hessian computations, resulting in high costs that scale poorly with model size. In contrast, `Emb` and `For-Value` are training-free and algorithm-agnostic, which significantly reduces overhead. Although `HyperINF` is the strongest baseline in terms of accuracy, its cubic complexity makes it impractical for large LLMs—requiring about 6 hours for a Qwen-32B model (Fig. 4b). Although `Emb` achieves the best runtime efficiency, its performance lags behind other methods, as demonstrated in Tab. 1 and Tab. 2. Our method, `For-Value`, maintains strong performance while remaining highly efficient. Since $|\hat{\mathcal{V}}|$ is typically small (often under 2k), `For-Value` achieves much lower computational and memory costs than baselines.

| Method | Training Free | Algorithm Agnostic | Training Complexity | Computational Complexity | Memory Complexity |
|---|---|---|---|---|---|
| Original IF | ✗ | - | $O(nEd_{in}dL)$ | $O(nd_{in}^2d^2L + d_{in}^3d^3L)$ | $O(D^2L + nDL)$ |
| Hessian-free | ✗ | ✗ | $O(nEd_{in}dL)$ | $O(nd_{in}dL)$ | $O(nd_{in}dL)$ |
| DataInf | ✗ | ✗ | $O(nEd_{in}dL)$ | $O(nd_{in}dL)$ | $O(nd_{in}dL)$ |
| HyperINF | ✗ | ✗ | $O(nEd_{in}dL)$ | $O(nd^3L)$ | $O(nd^2L)$ |
| Emb | ✓ | ✓ | 0 | $O(nd)$ | $O(nd)$ |
| For-Value (ours) | ✓ | ✓ | 0 | $O(nd|\hat{\mathcal{V}}|)$ | $O(nd|\hat{\mathcal{V}}|)$ |

Table 6: Comparison on complexity of the Influence Function (IF), `Hessian-free`, `DataInf`, `Emb`, and `For-Value`. Complexities are given assuming a multilayer perceptron (MLP) with $L$ layers, each containing $d_{in} \times d$ neurons where $d_{in}$ is input dimension and $d$ is the output embedding dimension, trained for $E$ epochs on $n$ training samples. The parameter count is identical across layers ($D \in \mathbb{N}$), and the in-batch volcabulary size is $|\hat{\mathcal{V}}|$. Overall, `For-Value` achieves higher computational and memory efficiency than baseline methods.

**Discussion on Parallel Computing:** While previous studies focus on using a single GPU for fair comparison, we would like to highlight that `For-Value` can further improve efficiency through parallel computing with a large batch size, as it only requires forward calculations. In contrast, baseline methods require computing the gradient for each individual data sample, which restricts them to a batch size of one and makes scaling up challenging.

**Qualitative Demonstration.** Beyond quantitative results, we present qualitative examples identified by `For-Value`. Fig. 5 shows a target valuation sample alongside its most and least influential training samples as ranked by `For-Value`. Specifically, `For-Value` successfully identifies highly relevant training points — for example, selecting samples from the same reverse order of words task for sentence transformation, or matching the same subject or artistic style in image-to-text tasks. In contrast, the least influential samples are clearly less relevant and often differ entirely in task or content from the target valuation data.

## A.5 ADDITIONAL DETAILS OF SELECT DATA FOR FINETUNING

**Mathematics: GSM8K** As the baseline methods require LoRA, we begin with a one-epoch warmup training on Llama3-8B Meta (2024) using the whole training set to avoid utilizing gradients from randomly initialized LoRA modules (with a rank of $r = 32$). Next, we calculate influence scores

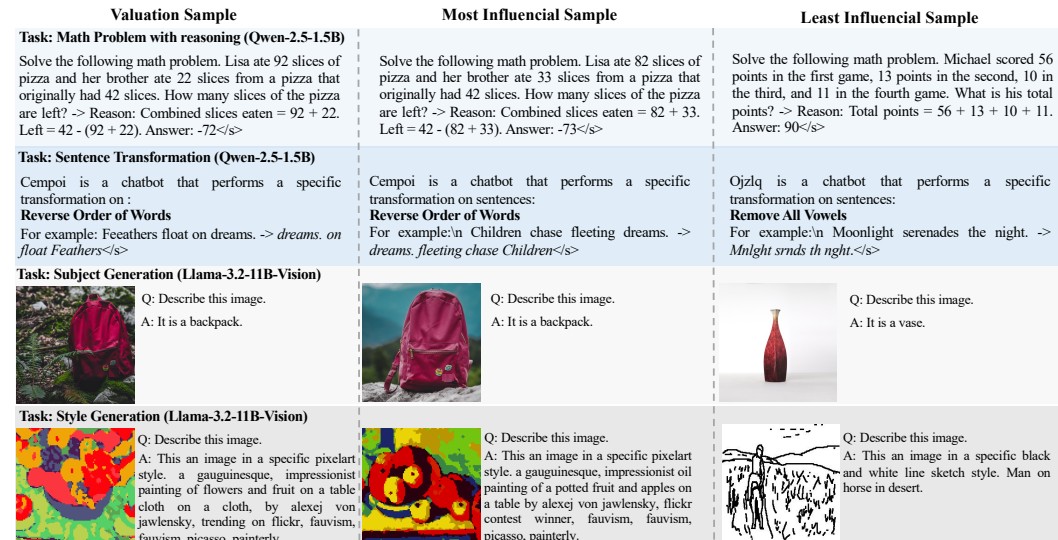

Figure 5: Qualitative examples of data influence identified by `For-Value`. For each target valuation sample (left column), the most influential (middle column) and least influential (right column) training samples are shown. `For-Value` correctly retrieves training samples that share relevant task characteristics (e.g., same reasoning type, sentence transformation rule, subject, or style) and filters out unrelated or mismatched examples.

for both the baselines and `For-Value`. To ensure consistency and performance, we also perform a one-epoch warm-up but with full-parameter finetuning on the entire dataset. Finally, we select the top 5% of data based on these influence scores to further finetune the model with learning rate $1e-5$ and batch size 64 on 4 H100 GPU for 4 epochs.

**Medicine: Noise-Huatuo-Complex-CoT** As the baseline methods utilize LoRA, we begin with a one-epoch training on Llama3-8B-Instruction Meta (2024) using the whole training set to avoid using gradients from randomly initialized LoRA modules (with a rank of $r = 16$). Next, we calculate influence scores for both the baselines and our approach. Considering the training data is noisy, we select the top 5% high value training data based on these scores and finetune the original pretrained model using full-parameter finetuning for 5 epochs, with a learning rate of $1 \times 10^{-6}$, a batch size of 16 and gradient accumulation 8 on 8 H100 GPUs. We follow Wu et al. (2025) using greedy decoding to evaluate the model on 5 held out datasets MedQA Jin et al. (2021), MedMCQA Pal et al. (2022), PubMedQA Jin et al. (2019), MMLU-Pro-Med Wang et al. (2024b), GPQA-Med Rein et al. (2024).

**Medicine: Noise-Huatuo-Complex-CoT** Similarly, we start with a one-epoch warm-up on the entire training set to prevent using gradients from randomly initialized LoRA modules (with a rank of $r = 16$). Then, we compute influence scores for the baseline methods. For our method, since the pretrained model already demonstrates sufficient medical knowledge (as shown by adequate test accuracy in Table 2), we directly use the original pretrained model to assess data value. Finally, we finetune the pretrained Qwen2.5-3B-VL model Bai et al. (2025a) with full-parameter finetuning for 3 epochs, using a learning rate of $1 \times 10^{-5}$, a batch size of 16, and gradient accumulation of 8 on 8 H100 GPUs. We evaluate the model with greedy decoding on 6 held out datasets: PMC Zhang et al. (2023), MMMU Yue et al. (2024), MedX-M Zuo et al. (2025), PathVQA He et al. (2020), SLAKE Liu et al. (2021), VQA-Rad Lau et al. (2018).

## A.6 ADDITIONAL ANALYSIS ON SELECT DATA FOR FINETUNING

**Medicine: Noise-Huatuo-Complex-CoT**. As indicated in Tab. 4, baseline methods struggle to effectively select high-quality data from noisy training datasets. This is primarily because these methods rely on assumptions of uniqueness or convergence to an optimal solution Bae et al. (2024), which are difficult to satisfy in the presence of noisy data. To illustrate this, we evaluated the proportion of high-quality data within the top 10% of high-value data, as shown in Tab. 7. The results reveal

| Llama-3.1-8B | Detection Accuracy |
|---|---|
| `Hessian-free` | 48.2 |
| `HyperINF` | 15.1 |
| `DataInf` | 33.2 |
| `For-Value` | 84.4 |

Table 7: High quality data detection accuracy

that baseline methods generally lack the capability to accurately identify noisy data, whereas our proposed method (`For-Value`) achieves significantly higher accuracy in detecting clean data.

Table 8: Description of the sentence transformation task templates. We consider 10 different types of sentence transformations. For each sentence transformation, unique identifying "chatbot" names were additionally prepended to the task prompt to assist the model in training.

| Sentence transformations | Example transformation of "Sunrises herald hopeful tomorrows": |
|---|---|
| Reverse Order of Words | tomorrows. hopeful herald Sunrises |
| Capitalize Every Other Letter | sUnRiSeS hErAlD hOpEfUl tOmOrRoWs. |
| Insert Number 1 Between Every Word | Sunrises 1herald 1hopeful 1tomorrows. |
| Replace Vowels with * | S*nr*s*s h*r*ld h*p*f*l t*m*rr*ws. |
| Double Every Consonant | SSunrriisseess hheraldd hhopefull ttomorrows. |
| Capitalize Every Word | Sunrises Herald Hopeful Tomorrows. |
| Remove All Vowels | Snrss hrld hpfl tmrrws. |
| Add 'ly' To End of Each Word | Sunrisesly heraldly hopefully tomorrows.ly |
| Remove All Consonants | uie ea oeu ooo. |
| Repeat Each Word Twice | Sunrises Sunrises herald herald hopeful hopeful tomorrows. tomorrows. |

## A.7 DETAILED TASK DESCRIPTION

### A.7.1 LLM INFLUENCE EVALUATION TASKS

Following (Kwon et al., 2024), we evaluate the performance of `For-Value` on three text generation tasks for large language models (LLMs) to identify influential data points:

- **Sentence Transformations:** This task requires transforming input sentences into alternative forms while preserving meaning (e.g., active to passive voice). The dataset comprises 10 distinct classes (e.g., declarative to interrogative), each with 100 examples, split into 90 training and 10 test examples per class. Data examples see Tab. 8.

- **Math Word Problems (Without Reasoning):** These problems involve direct numerical computation from textual descriptions (e.g., basic arithmetic). The dataset has 10 classes based on operation types, with 100 examples per class (90 training, 10 test). Data examples see Tab. 9.

- **Math Word Problems (With Reasoning):** These require multi-step reasoning (e.g., solving word problems involving algebra or logic). Similar to the previous task, the dataset includes 10 classes with 100 examples each (90 training, 10). Data examples see Tab. 9.

### A.7.2 VLM INFLUENCE EVALUATION TASKS

For VLMs, we adapt text-to-image generation tasks from (Kwon et al., 2024) into image-to-text (captioning) tasks to evaluate influence:

- **Style Generation:** This task involves generating captions for images in specific styles: cartoons (Norod78, 2023), pixel art (Jainr3, 2023), and line sketches (Zoheb, 2023). Each style dataset contains 200 training and 50 test image-text pairs, totaling 600 training and 150 test samples across three styles. Data examples see Fig. 5.

- **Subject Generation:** Using the DreamBooth dataset (Ruiz et al., 2023), this task generates captions for images of 30 distinct subjects (e.g., specific objects or animals). Each subject provides 3 training samples, with the remaining samples used for valuation. Data examples see Fig. 5.

Table 9: Description of the math problem task templates. We consider 10 different types of math word problems.

| Math Word Problems | Template prompt question |
|---|---|
| Remaining pizza slices | Lisa ate A slices of pizza and her brother ate B slices from a pizza that originally had C slices. How many slices of the pizza are left? Reason: Combined slices eaten = A + B. Left = C - (A + B). |
| Chaperones needed for trip | For every A students going on a field trip, there are B adults needed as chaperones. If C students are attending, how many adults are needed? Reason: Adults needed = (B * C) // A. |
| Total number after purchase | In an aquarium, there are A sharks and B dolphins. If they bought C more sharks, how many sharks would be there in total? Reason: Total sharks = A + C. |
| Total game points | Michael scored A points in the first game, B points in the second, C in the third, and D in the fourth game. What is his total points? Reason: Total points = A + B + C + D. |
| Total reading hours | Emily reads for A hours each day. How many hours does she read in total in B days? Reason: Total hours read = A * B. |
| Shirt cost after discount | A shirt costs A. There's a B-dollar off sale. How much does the shirt cost after the discount? Reason: Cost after discount = A - B. |
| Area of a garden | A rectangular garden has a length of A meters and a width of B meters. What is its area? Reason: Area = A * B. |
| Total savings | If Jake saves A each week, how much will he save after B weeks? Reason: Total savings = A * B. |
| Number of cupcake boxes | A bakery sells cupcakes in boxes of A. If they have B cupcakes, how many boxes can they fill? Reason: Boxes filled = B // A. |
| Interest earned | John invests A at an annual interest rate of B%. How much interest will he earn after C years? Reason: Interest = (A * B * C) // 100. |

### A.7.3 INFLUENTIAL DATA DETECTION METRICS

We adopt two metrics from (Kwon et al., 2024) to assess influence:

- **AUC Score:** For each test data point, we assign pseudo labels to training points (1 if the training point's label matches the test point's, 0 otherwise). We compute the Area Under the Curve (AUC) between data values (influence scores) and pseudo labels, averaging across all test points. A higher AUC indicates better identification of influential points.

- **Recall:** For each test point, we calculate the percentage of influential training points (top-ranked by influence score) that share the same class as the test point. This measures the relevance of identified influential points.

### A.7.4 MISLABELED DATA DETECTION DATA & METRICS

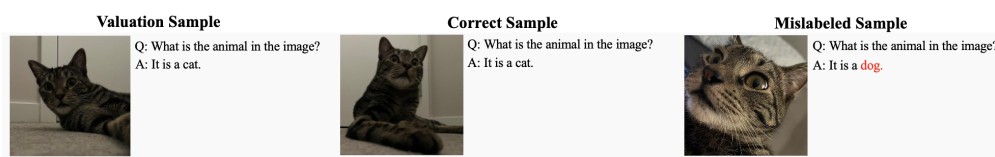

Figure 6: Description of the mislabeled data detection task. We utilize a cat versus dog classification dataset and intentionally introduce noise by randomly swapping the labels of 50% of the data.

For mislabeled detection, we transform the dataset into a visual-language question answering task with the template "What is the animal in the image? It is a [label]" with demonstration[3] in Fig. 6. We then select the first 400 images for both dogs and cats, flipping 50% of the labels to introduce noise. For valuation, we use 200 images, with each class containing 100 images. For evaluation, we also calculate the AUC and Recall but with the pseudo labels to training points being 1 if the training point's label matches the test point's and it is clean data, 0 otherwise.

## A.8 NOISE-HUATUO-CoT DATA EXAMPLE

We construct the Noise-Huatuo-Complex-CoT dataset by randomly sampling 5,000 examples without replacement and injecting noise into 40% of them through random insertion or deletion of irrelevant words, as illustrated in Fig. 7.

| Random Drop | Random Insert |
| --- | --- |
| **Question:** A nucleic acid was analyzed and found to contain 32% adenine, 18% guanine, 17% cytosine and 33deg/o thymine. The nucleic acid must be: A. Single-stranded RNA B. Single-stranded DNA C. Double-stranded RNA D. Double stranded DNA | **Question:** What physiological process in the collecting tubules is most likely to be decreased upon administration of a drug that blocks epithelial sodium channels in the cortical collecting duct? |
| ## Alright, let's figure out what kind of nucleic acid we're with here. that's adenine, guanine, 17% cytosine, and 33% first things first, in double-stranded DNA, you'd expect adenine to pair with pair with cytosine. That means their should looking at is 32% while really close. Guanine is 18% and 17%. close. This of makes me think of double-stranded DNA because that's where see this kind of pairing. wait, not jump conclusions too quickly. Single-stranded DNA doesn't really care about matching percentages, so technically, it could have any mix of bases. be single-stranded DNA? It's possible, since there's no rule the strands. Now, out instead of at 33%. If it were RNA, we'd be seeing instead, So, it can't be neither We're definitely dealing with DNA here. Hmm, circling those matching percentages of A and T, G and really at a characteristic balance. show was a reason for such balance, which seems unlikely here without any it. considering all of this, the nucleic acid we're looking DNA. The complementary just too that seems to make the most sense given we've got. ## D. Double-stranded DNA. double-stranded Additionally, presence of thymine rather uracil rules out any RNA options. | ## Thinking bar bar Alright, let's think about these epithelial sodium bar foo bar bar channels. They're bar sitting there in the foo foo apical membrane of those epithelial cells – specifically in the cortical collecting duct foo baz foo bar foo – doing the job of pulling sodium foo bar foo bar foo bar bar from the bar tubular fluid back foo baz into bar the bloodstream. bar Super important foo for foo baz bar sodium balance, right? So, if we throw a wrench in that foo system by blocking these channels, what happens? Hmm, yeah, sodium can't baz baz bar get reabsorbed, that's for sure. Less sodium going back into the blood – that's foo baz foo significant! Now, let's consider the water side foo of things. **...** of a drug that foo bar bar blocks epithelial sodium channels in baz the cortical collecting duct is sodium reabsorption. When these channels foo foo are inhibited, sodium cannot be reabsorbed from the tubular fluid back into the bloodstream. Consequently, this disruption also affects water bar foo bar reabsorption due to the osmotic relationship between bar baz sodium baz and water, and it can alter potassium handling, although foo the primary and most immediate effect is on baz baz sodium reabsorption. |

Figure 7: Examples of two types of noisy data. (Left) Random word deletion, where tokens are dropped from the reasoning, for instance, 'Thinking' is removed after ##. (Right) Random word insertion, where irrelevant tokens such as 'bar,' 'foo,' and 'baz' are injected into the reasoning. Red dashes means omitted reasoning.

### A.8.1 BASELINE CHECKPOINTS SELECTION

For baseline methods, we select the model checkpoint with the highest test AUC, as influence function-based methods exhibit significant performance variability across training checkpoints. Notably, this variability does not correlate with validation loss, posing challenges for practical deployment. We compare `For-Value` against these baselines to ensure robust evaluation.

### A.8.2 DATASET STATISTICS

We present dataset statistics in Tab. 10

## A.9 USAGE OF LARGE LANGUAGE MODEL

In preparing this paper, we made limited use of ChatGPT to support writing and editing. Specifically, LLMs were employed for language polishing, grammar refinement, and rephrasing sentences to improve clarity and readability. Importantly, all technical content, including theoretical analysis, algorithm design, and experimental results, was conceived, implemented, and validated by the

---

[3]To prevent any licensing issues, the images shown are not from the original dataset; they were personally captured for demonstration purposes.

Table 10: Dataset statistics for LLM and VLM tasks.

| Task | Training Samples | Valuation Samples |
|------|------------------|-------------------|
| Sentence Transformations | 900 (90 × 10 classes) | 100 (10 × 10 classes) |
| Math Word Problems (No Reasoning) | 900 (90 × 10 classes) | 100 (10 × 10 classes) |
| Math Word Problems (With Reasoning) | 900 (90 × 10 classes) | 100 (10 × 10 classes) |
| Style Generation | 600 (200 × 3 styles) | 150 (50 × 3 styles) |
| Subject Generation | 90 (3 × 30 subjects) | Variable (1-3) per subject |
| Mislabel Detection | 800 (400 × 2 subjects 50% noise) | 200 (100 × 2 subjects) |
| GSM8K | 7470 | 1319 |
| Noise-Huatuo-Complex-CoT | 5000 (2981 clean, 2019 noise) | 5000 (clean) |
| PMC-Reasoning (subset) | 10000 | 5000 |

authors. LLM outputs were always critically reviewed, verified, and revised before inclusion. No LLM-generated text, figures, or tables were incorporated without careful human oversight.

## A.10  LICENSE CLARIFICATION

The Dreambooth images have been either taken by the authors of the paper or obtained from Unsplash[4]. The file located at this link[5] includes a list of all reference links to the images on Unsplash, along with the photographers' attributions and the image licenses. The sketch images are sourced from FS-COCO Chowdhury et al. (2022). Data attributions and image licenses can be found in the file provided at the following link[6].

---

[4]https://www.unsplash.com/

[5]https://huggingface.co/datasets/google/dreambooth/blob/main/dataset/references_and_licenses.txt

[6]https://github.com/pinakinathc/fscoco

