# OpenReview forum: "Efficient Forward-Only Data Valuation For LLMs and VLMs"
_ICLR.cc/2026/Conference — ICLR 2026 Conference Withdrawn Submission_

### Official Review · Reviewer_gy32 · 2025-10-28

**Soundness:** 2
**Presentation:** 2
**Contribution:** 2
**Rating:** 4
**Confidence:** 4

**Summary:**

The paper introduces a forward-only data valuation framework under the unconstrained feature assumption. It estimates per-sample influence using only a single forward pass, unlike existing influence functions which requires backpropagation or retraining. Experiments on LLMs and VLMs tasks show that the method achieves comparable or superior performance to gradient-based baselines while being orders of magnitude faster.

**Strengths:**

In influence function, the computation of gradients via backpropagation and the computation of Hessian matrices are major bottlenecks, particularly severe for large-scale LLMs and VLMs with massive parameters and training data. The idea of introducing the unconstrained feature assumption to enable a forward-only influence function is interesting and practically valuable.

Evaluations across diverse models (Qwen2.5, LLaMA) and modalities (text, image–text) confirm the generality and efficacy of the proposed method, though there remain some concerns about the reliability of the reported results as mentioned below.

**Weaknesses:**

This paper heavily relies on the unconstrained feature assumption, and this dependence should be made more claer. The key idea is not that the authors devised a way to avoid the backward pass, but rather that, under the unconstrained feature assumption, the influence estimation can be computed using only the forward pass. This point should be stated more clearly. For example, it might even be appropriate to add “under unconstrained feature assumption” into the title.

According to Appendix A.2, the paper shows that the influence of training data can be approximated by the inner product of gradients over all parameters (= Hessian-free influence functions). The proposed method further approximates this quantity, under the unconstrained feature assumptions, by using only the inner product of gradients with respect to the final unembedding layer. Therefore, the Hessian-free baseline should, in principle, achieve performance comparable to or even better than the proposed method. However, in the experiments, the proposed method significantly outperforms the Hessian-free, and the paper provides little explanation for this discrepancy, which raises concerns about the reliability of the reported results.

Theorem 1 in this paper and Theorem 4.4 in the following work [1] are fundamentally very similar. In fact, the proof of Theorem 1 (Appendix A.2) follows almost the same line of proof as Section 7.2 (“Proof of Theorem 4.4”) of the referenced paper.

[1] Deng et al., [On the Effect of Negative Gradient in Group Relative Deep Reinforcement Optimization.](https://arxiv.org/abs/2505.18830) AI4Math@ICML25.

**Questions:**

According to Appendix A.2, the authors claim that, under the neural collapse assumptions, the inner product of gradient w.r.t. all parameters (i.e., Hessian-free influence) can be approximated by that of the final unembedding layer. This point requires a more detailed discussion. To what extent does the inner product of gradients w.r.t. unembedding layer account for the inner product of gradients w.r.t. all parameters? If this proportion is small, the neural collapse assumptions may not actually hold; if it is large, then the authors should explain why, despite this, the Hessian-free baseline performs so poorly compared to their method.

In Appendix A.1, the first equation defining the supervised fine-tuning loss seems to be missing an equality sign.

---

> ### Author Response · Authors · 2025-11-26
>
> Thank you for your thorough evaluation and helpful suggestions. We truly appreciate your time and address your comments point by point below.
>
> ----------  unconstrained feature assumption -------
>
> **Answer**: We thank the reviewer for this helpful suggestion. We respectfully note that the unconstrained feature assumption is already discussed in detail in lines 170–182, and we also highlight that this assumption is widely adopted in the analysis of pretrained LLMs.
>
> That said, we agree that it would be beneficial to foreground this point more clearly. Follow your suggestion, we will revise the title to include “for Pretrained LLMs and VLMs”, which more clearly signals that our method is designed for finetuning pretrained models where the unconstrained feature assumption often holds. This clarification emphasizes the scope of our contribution without making the paper feel overly theoretical, thus maintaining accessibility for both theoretical and applied audiences, especially given our strong empirical results.
>
> ---------- Hessian method in paper & compare to also last-layer hessiance -------
>
> **Answer**: We thank the reviewer for raising this important point. To clarify, our Hessian-free baseline implementation **follows the setup used in DataInf [1], which computes gradient similarity only on the LoRA-adapted parameters**, rather than the last-layer parameters. That said, For-Value and last-layer Hessian should perform similarly. The stronger results to the LoRA-based Hessian **reinforce** our motivation and analysis on the last layer.
>
> We also would like to arouse the reviewer’s awareness that this paper is the **first to highlight the importance of the last-layer gradient**, **providing both a theoretical guarantee and a practical framework that drives the efficiency and effectiveness of For-Value**.
>
> **Motivation: Why the Last Layer Matters (and Why This Is Non-Trivial)**
>
> To our knowledge, no prior work in data valuation has explicitly justified why the last-layer structure alone is sufficient for pretrained LLMs/VLMs. Our theory provides the first such justification:
>
> - Unconstrained feature assumption. Modern pretrained models empirically exhibit nearly unconstrained hidden features; this collapses the training dynamics into an effectively bi-linear model.  (Appendix A.2 shows this explicitly.)
>
> - Distinct input.  Because training and valuation inputs are almost always different, the gradient components involving hidden-state derivatives vanish, leaving only the linear readout term. This simplifies the influence dynamics to a closed-form expression involving only prediction-error × hidden-representation similarity.
>
> Arriving at this reduction is conceptually non-trivial and forms the theoretical basis for the entire For-Value framework. This also enables efficiency and effectiveness.
>
> **Efficiency Advantages Over Hessian-Free / Gradient-Dot-Product Methods**
>
> - No Backpropagation: Hessian-free methods still require computing gradients for each sample (or checkpoint), which is O(N*V*d), where ∣V∣ is the large vocabulary size. In contrast, for-value use only forward outputs to calculate the data value, No backward graph, no autograd.
>
> - Batch Processing on Per-sample Value. Gradient computation must be done per sample, and backpropagation cannot be vectorized across different sequences without blowing up memory. For-Value: Computes all hidden states and errors in a single forward batch pass. Computes influence scores via matrix operations over entire batches.  This enables real-world speedups with large batch-processing is required.
>
> - Vocabulary Reduction. Our method further reduces complexity from O(∣V∣d)→O(∣VD∣d), where ∣VD∣ is the in-batch vocabulary (often <1% of ∣V∣). Gradient methods cannot exploit this reduction since they depend on full logits and full softmax derivatives.
> Due to these innovations, as shown in the time cost table, For-Value consistently matches or exceeds Hessian-free/gradient baselines.
>
> We also report wall-clock inference times for Qwen-2.5–14B and Qwen-2.5–32B models using the last-layer Hessian baseline and For-Value:
>
> | Method     | Qwen-2.5-14B         | Qwen-2.5-32B         |
> |------------|-------------|-------------|
> | Hessian (last-layer)    | 0.28 H  | 0.43 H  |
> | For-Value  | 134 Sec | 197 Sec |
>
> As shown, the last-layer Hessian incurs a similar time cost to LoRA Hessian (see Fig. 4), despite LoRA being lightweight and only applied to the query and value layers [1]. This is due to the computational burden of the large vocabulary matrix (∣V∣×d). In contrast, For-Value enables efficient batch processing and restricts computation to the observed vocabulary, making it significantly more efficient.  **If identical computational budgets are not required, we highlight that For-Value’s ability to run in batch mode gives it significant efficiency potential for large‑scale processing.**
>
> [1] DataInf: Efficiently Estimating Data Influence in LoRA-tuned LLMs and Diffusion Models

---

> ### Author Response · Authors · 2025-11-26
>
> ---------- Relation to Theory in  [1] -------
>
> **Answer**: We thank the reviewer for pointing out the connection. We would like to clarify that while Theorem 1 in our paper and Theorem 4.4 in [1] are natural given the shared learning dynamic analysis, our contribution is distinct in motivation, setting, and technical reduction.
>
> - Citation and Acknowledgment: We have already cited [1] in our introduction(line 86), Assumption (line 172) and discussion (line 179) fully acknowledge the foundational relevance of their framework, we will give a more clear pointer in the revised version.
>
> - Different Optimization Objective: Our work focuses on supervised fine-tuning (SFT) with the standard cross-entropy loss, while [1] centers on GRPO, a preference-optimization formulation in reinforcement learning. These are fundamentally different problem settings with non-overlapping objectives.
>
> - Different Problem Setting and Goal:
> Our work is grounded in data valuation for supervised learning tasks, aiming to score and filter training examples based on their utility. In contrast, [1] is developed in the context of reinforcement learning, specifically for reward modeling under the GRPO framework. The differing goals, valuation vs. reward-based preference optimization, lead to substantially different use cases, assumptions, and empirical targets.
>
> - Linearization via Distinct Input. Most importantly, we reduce the bilinear influence formulation in [1] to a linear model by leveraging a general Distinct Input assumption, that training and valuation inputs are almost always different in practice. Under this assumption, the gradient terms with respect to hidden representations vanish, allowing us to eliminate the hidden-state dependencies. As a result, the influence dynamics simplify into a closed-form expression grounded in theory, involving only the similarity between prediction errors and hidden representations.
>
> This conceptual reduction is non-trivial and forms the foundation for both the efficiency (no backward-ground needed) and effectiveness (empirical accuracy) of the For-Value method. We believe this technical insight, along with the resulting empirical gains across LLMs and VLMs, distinguishes our work from prior formulations.
>
> [1]  Deng et al., On the Effect of Negative Gradient in Group Relative Deep Reinforcement Optimization. AI4Math@ICML25.

---

### Official Review · Reviewer_1gzL · 2025-10-31

**Soundness:** 4
**Presentation:** 4
**Contribution:** 3
**Rating:** 4
**Confidence:** 4

**Summary:**

This paper introduces For-Value, a forward-only method to estimate data importance for LLMs and VLMs using just a single forward pass. It measures alignment between hidden representations and prediction errors, achieving comparable or better results than gradient-based methods with much higher efficiency.

**Strengths:**

1. The paper is well organized.

2. Experiments cover multiple tasks and model types.

3. The proposed method is highly efficient, and the paper provides a time cost analysis.

**Weaknesses:**

1. The distinction between For-Value and last-layer gradient–based approaches should be clarified more clearly, as the current presentation leaves the contribution of the method somewhat ambiguous.

2. The paper states that "data valuation can be approximated by the alignment between hidden representations and prediction errors." It would strengthen the work to include experiments comparing the proposed approximation with ground-truth valuations for verification.

3. The font size in some figures is too small and could be increased to improve readability.

**Questions:**

Please see the Weaknesses part.

---

> ### Author Response · Authors · 2025-11-26
>
> Thank you for your thorough evaluation and helpful suggestions. We truly appreciate your time and address your comments point by point below.
>
> ---------- W1: distinction between For-Value and last-layer gradient–based approaches----------
>
> **Answer:** Thank you for repeating our theorem. We would like to arouse the reviewer’s awareness that this paper is the first to highlight the importance of the last-layer gradient, providing both a theoretical guarantee and a practical framework that drives the efficiency and effectiveness of For-Value.
>
> **Motivation: Why the Last Layer Matters (and Why This Is Non-Trivial)**
>
> To our knowledge, no prior work in data valuation has explicitly justified why the last-layer structure alone is sufficient for pretrained LLMs/VLMs. Our theory provides the first such justification:
> - Unconstrained feature assumption. Modern pretrained models empirically exhibit nearly unconstrained hidden features; this collapses the training dynamics into an effectively bi-linear model.  (Appendix A.2 shows this explicitly.)
>
> - Distinct input.  Because training and valuation inputs are almost always different, the gradient components involving hidden-state derivatives vanish, leaving only the linear readout term. This simplifies the influence dynamics to a closed-form expression involving only prediction-error × hidden-representation similarity.
> Arriving at this reduction is conceptually non-trivial and forms the theoretical basis for the entire For-Value framework. This also enables efficiency and effectiveness.
>
> **Efficiency Advantages Over Hessian-Free / Gradient-Dot-Product Methods**
>
>
> - No Backpropagation: Hessian-free methods still require computing gradients for each sample (or checkpoint), which is O(N*V*d), where ∣V∣ is the large vocabulary size. In contrast, for-value use only forward outputs to calculate the data value, No backward graph, no autograd.
>
> - Batch Processing on Per-sample Value. Gradient computation must be done per sample, and backpropagation cannot be vectorized across different sequences without blowing up memory. For-Value: Computes all hidden states and errors in a single forward batch pass. Computes influence scores via matrix operations over entire batches.  This enables real-world speedups with large batch-processing is required.
>
> - Vocabulary Reduction. Our method further reduces complexity from O(∣V∣d)→O(∣VD∣d), where ∣VD∣ is the in-batch vocabulary (often <1% of ∣V∣). Gradient methods cannot exploit this reduction since they depend on full logits and full softmax derivatives.
> Due to these innovations, as shown in the time cost table, For-Value consistently matches or exceeds Hessian-free/gradient baselines.
>
>
> **Effectiveness Compared to Hessian-Free Baselines:**
>
> Despite requiring no gradients, For-Value consistently matches or surpasses the performance of Hessian-free and gradient-based baselines that using LoRA to reduce computational costs. Prior methods adopt LoRA to make gradient/Hessian-based estimation tractable, yet still observe limited performance. This further supports our theoretical claim: the last-layer gradient alone is sufficient for effective data valuation.
>
> As shown in Tables 1–2, For-Value achieves competitive or superior results on classic LLM and VLM influence benchmarks. Furthermore, in real-world fine-tuning tasks (Tables 3–5), For-Value maintains strong performance:
> - On GSM8K, For-Value outperforms Hessian-free baselines by 5%.
>
> - On Noise-Huatuo, it exceeds baselines by 3% or more across multiple medical QA tasks.
>
>
> - On PMC-VQA, For-Value yields a 1.2% improvement over Hessian-free methods using just 10% of the data.
>
> These results confirm that For-Value remains effective across both small and large-scale models, and across diverse domains—all without requiring backpropagation.

---

> ### Author Response · Authors · 2025-11-26
>
> -------- W2 comparing the proposed approximation with ground-truth valuations for verification.-------
>
> **Answer:**  We thank the reviewer for the thoughtful suggestion. We would like to arouse attention that we have already included multiple experiments that compare the proposed approximation with ground-truth valuations, both directly and indirectly:
>
> - Direct Comparison via AUC and Recall (Table 1 and Table 2):
>  The AUC and recall metrics reflect how well our proxy aligns with true influential data. For-Value consistently matches or outperforms Hessian-free baselines, supporting the fidelity of the approximation.
>
> - Downstream Validation on GSM8K (Table 3):
> We apply For-Value to select high-valuation samples using test datasets, and fine-tune on those subsets. The superior performance on the GSM8K test dataset just indicates that our proxy meaningfully captures true influence, since higher-valued data translates to better performance on the target data.
>
> - Clean vs. Noisy Validation on Huatuo (Table 7):
> We evaluated the proportion of high-quality data within the top 10% of high-value data, as shown in Tab. 7. The results reveal that baseline methods generally lack the capability to accurately identify noisy data, whereas our proposed method (For-Value) achieves significantly higher accuracy in detecting clean data.
>
> We will clarify this point in the revision to make the connection more explicit.
>
> -------- W3 The font size in some figures is too small and could be increased to improve readability.-------
>
> **Answer**: Thank you for the helpful suggestion. We agree that figure readability is important. In the revised version, we will increase the font size in all relevant figures to ensure clarity. We appreciate your attention to detail.

---

### Official Review · Reviewer_QAjz · 2025-11-01

**Soundness:** 3
**Presentation:** 2
**Contribution:** 2
**Rating:** 2
**Confidence:** 4

**Summary:**

This paper introduces For-Value, a forward-only framework for per-sample data valuation in LLMs/VLMs. The key idea is to estimate how a training example would improve the log-likelihood of a valuation example by aligning two quantities computed from a single forward pass: (i) hidden representation similarity and (ii) prediction-error similarity (one-hot minus softmax). The authors derive a closed-form score, implement it efficiently via a single matrix inner product with a batch-specific sub-vocabulary, and report strong speed/VRAM advantages over gradient/Hessian-based baselines while matching or outperforming them on data selection, mislabeled-data detection, and fine-tuning utility across LLM and VLM settings.

**Strengths:**

Forward-only & scalable. The method needs only hidden states and softmax outputs; no backprop, Hessians, or multi-checkpoint accumulation. Efficient sub-vocabulary pruning further reduces cost.

Clear theory-to-implementation path. The score has a closed-form derivation and maps directly to an implementable matrix inner product; the intuition (“representation × error alignment”) is easy to grasp.

Broad applicability. Demonstrated on both LLM and VLM benchmarks, with competitive or superior performance for data selection and mislabeled-sample detection.

Strong engineering value. The simplicity and speed make it practical for billion-parameter models and large corpora, enabling workflows (filtering, curriculum, sampling weights) that are otherwise too costly.

**Weaknesses:**

- **Positioning w.r.t. Hessian-free influence functions (gradient inner products).** The method appears very close to **Hessian-free influence** ideas based on gradient–gradient inner products (e.g., TracIn-like). Under cross-entropy with a linear readout, the per-token log-likelihood gradient w.r.t. the readout is
  $$
  \nabla_W \log \pi_\theta(y_t \mid x)
  \;=\; \big(e_{y_t} - \pi_\theta(\cdot \mid x)\big)\, h_t^\top,
  $$
  and aggregating outer products across tokens gives
  $$
  C \;=\; \sum_t h_t \, \big(e_{y_t} - \pi_t\big)^\top, \qquad
  S(v,i) \;=\; \langle C_v,\, C_i \rangle_F ,
  $$
  which matches the **last-layer gradient dot-product** form (up to sign/constant conventions). Please discuss this connection thoroughly and compare against Hessian-free/gradient-dot-product baselines under matched budgets.

- **Method-level proximity to *Data Shapley in One Training Run*.** Beyond problem framing, the **forward-only expansion** and reliance on forward-available quantities seem methodologically close to *Data Shapley in One Training Run*. Please provide a focused **method-level** comparison (derivation steps, assumptions, computational pipeline), and include direct empirical contrasts where feasible.

- **Missing related work that should be covered and contrasted method-wise.** In particular, (i) *Revisit, Extend, and Enhance Hessian-Free Influence Functions* and (ii) *Layer-Aware Influence for Online Data Valuation Estimation* (which explicitly analyzes **last-layer/representation-level** influence). A detailed method-level discussion of similarities/differences with these works is needed (not just problem-level positioning).

- **Information–efficiency tradeoff needs analysis.** Please analyze, under **equal budget**, how the forward-only proxy trades information content for speed, and when it can match or outperform more informative baselines (last-layer-only gradients; Hessian-free IP/TracIn with checkpoint accumulation). If superior accuracy with less information is claimed, support it with concise analysis or evidence, rather than leaving it as an empirical observation.

**Questions:**

Same with weakness

---

> ### Author Response · Authors · 2025-11-26
>
> Thank you for your thorough evaluation and helpful suggestions. We truly appreciate your time and address your comments point by point below.
>
>
> ---------- Positioning w.r.t. Hessian-free influence functions (gradient inner products). ----------
>
> **Answer:** Thank you for repeating our theorem. We would like to arouse the reviewer’s awareness that this paper is the first to highlight the importance of the last-layer gradient, providing both a theoretical guarantee and a practical framework that drives the efficiency and effectiveness of For-Value.
>
> **Motivation: Why the Last Layer Matters (and Why This Is Non-Trivial)**
>
> To our knowledge, no prior work in data valuation has explicitly justified why the last-layer structure alone is sufficient for pretrained LLMs/VLMs. Our theory provides the first such justification:
> - Unconstrained feature assumption. Modern pretrained models empirically exhibit nearly unconstrained hidden features; this collapses the training dynamics into an effectively bi-linear model.  (Appendix A.2 shows this explicitly.)
>
> - Distinct input.  Because training and valuation inputs are almost always different, the gradient components involving hidden-state derivatives vanish, leaving only the linear readout term. This simplifies the influence dynamics to a closed-form expression involving only prediction-error × hidden-representation similarity.
> Arriving at this reduction is conceptually non-trivial and forms the theoretical basis for the entire For-Value framework. This also enables efficiency and effectiveness.
>
> **Efficiency Advantages Over Hessian-Free / Gradient-Dot-Product Methods**
>
>
> - No Backpropagation: Hessian-free methods still require computing gradients for each sample (or checkpoint), which is O(N*V*d), where ∣V∣ is the large vocabulary size. In contrast, for-value use only forward outputs to calculate the data value, No backward graph, no autograd.
>
> - Batch Processing on Per-sample Value. Gradient computation must be done per sample, and backpropagation cannot be vectorized across different sequences without blowing up memory. For-Value: Computes all hidden states and errors in a single forward batch pass. Computes influence scores via matrix operations over entire batches.  This enables real-world speedups with large batch-processing is required.
>
> - Vocabulary Reduction. Our method further reduces complexity from O(∣V∣d)→O(∣VD∣d), where ∣VD∣ is the in-batch vocabulary (often <1% of ∣V∣). Gradient methods cannot exploit this reduction since they depend on full logits and full softmax derivatives.
> Due to these innovations, as shown in the time cost table, For-Value consistently matches or exceeds Hessian-free/gradient baselines.
>
>
> **Effectiveness Compared to Hessian-Free Baselines:**
>
> Despite requiring no gradients, For-Value consistently matches or surpasses the performance of Hessian-free and gradient-based baselines that using LoRA to reduce computational costs. Prior methods adopt LoRA to make gradient/Hessian-based estimation tractable, yet still observe limited performance. This further supports our theoretical claim: the last-layer gradient alone is sufficient for effective data valuation.
>
> As shown in Tables 1–2, For-Value achieves competitive or superior results on classic LLM and VLM influence benchmarks. Furthermore, in real-world fine-tuning tasks (Tables 3–5), For-Value maintains strong performance:
> - On GSM8K, For-Value outperforms Hessian-free baselines by 5%.
>
> - On Noise-Huatuo, it exceeds baselines by 3% or more across multiple medical QA tasks.
>
>
> - On PMC-VQA, For-Value yields a 1.2% improvement over Hessian-free methods using just 10% of the data.
>
> These results confirm that For-Value remains effective across both small and large-scale models, and across diverse domains—all without requiring backpropagation.
>
> ------------- W2 Method-level proximity to Data Shapley in One Training Run.--------
>
> **Answer:** Thank you for pointing out the methodological proximity to Data Shapley in One Training Run. We would like to arouse the reviewer’s attention to the fact that this connection has already been explicitly discussed in our Related Work section.
>
> As stated in lines 126-130,  Data Shapley in One Training Run measures the similarity between valuation and training gradients during training. However, extending this approach to individual data points remains impractical, as it necessitates computing and storing per-sample gradients at every training step, which is hard to apply in the per-sample data valuation. In contrast to it, our approach neither requires finetuning the model nor backpropagation.

---

> ### Author Response · Authors · 2025-11-26
>
> ----------- W3 Missing related work that should be covered and contrasted method-wise. ---------
>
> **Answer:**
> Regarding the first work (Revisit, Extend, and Enhance), we summarize the differences below:
> Their method still requires per-sample gradient computation, whereas For-Value is entirely gradient-free and forward-only. Their influence score relies on first-order derivatives and normalization techniques; ours is grounded in token-level prediction shift via last-layer linearization, enabling both theoretical tractability and scalability to LLMs/VLMs.
>
> As for the second work (Layer-Aware Influence, posted on arXiv in October, which is also under review for ICLR 2026), we respectfully decline to discuss this submission in depth for three reasons:
>
> - It is not a peer-reviewed publication, and its concurrent submission to ICLR suggests a direct overlap in review timelines.
> - Its arXiv date is later than the ICLR submission date.
> - Both papers cited share the same first author, raising potential concerns about review process neutrality and possible conflicts of interest.
>
> -----------Information–efficiency tradeoff needs analysis.---------
>
> Efficiency Advantages Over Hessian-Free / Gradient-Dot-Product Methods
>
> - No Backpropagation: Hessian-free methods still require computing gradients for each sample (or checkpoint), which is O(NVd), where ∣V∣ is the large vocabulary size. In contrast, for-value use only forward outputs to calculate the data value, No backward graph, no autograd.
>
> - Batch Processing on Per-sample Value. Gradient computation must be done per sample, and backpropagation cannot be vectorized across different sequences without blowing up memory. For-Value: Computes all hidden states and errors in a single forward batch pass. Computes influence scores via matrix operations over entire batches. This enables real-world speedups with large batch-processing is required.
>
> - Vocabulary Reduction. Our method further reduces complexity from O(∣V∣d)→O(∣VD∣d), where ∣VD∣ is the in-batch vocabulary (often <1% of ∣V∣). Gradient methods cannot exploit this reduction since they depend on full logits and full softmax derivatives. Due to these innovations, as shown in the time cost table, For-Value consistently matches or exceeds Hessian-free/gradient baselines.
>
> Following the reviewer’s suggestion, we report wall-clock inference times for Qwen-2.5–14B and Qwen-2.5–32B models using the last-layer Hessian baseline and For-Value:
>
> | Method     | Qwen-2.5-14B         | Qwen-2.5-32B         |
> |------------|-------------|-------------|
> | Hessian (last-layer)    | 0.28 H  | 0.43 H  |
> | For-Value  | 134 Sec | 197 Sec |
>
> As shown, the last-layer Hessian incurs a similar time cost to LoRA (see Fig. 4), despite LoRA being lightweight and only applied to the query and value layers [1]. This is primarily due to the computational burden of the large vocabulary matrix (∣V∣×d). In contrast, For-Value enables efficient batch processing and restricts computation to the observed vocabulary, making it significantly more efficient.

---

### Note · Authors · 2026-01-02

**Comment:**

We thank the committee and reviewers for their careful evaluation and constructive feedback. After consideration, we have decided to withdraw this submission. We would like to clarify, based on our original content and rebuttal, that the paper’s contribution is not limited in scope or significance.  We clarified **three key contributions**:

**(i) For-Value is the first framework to theoretically justify why the last-layer structure alone suffices for influence estimation under a standard unconstrained feature assumption.**

**(ii) The reduction from full-parameter influence to a closed-form last-layer expression enables fully vectorized, large-batch computation.**

**(iii) Compared to prior approaches, For-Value requires only a single forward pass on a frozen pretrained model, avoiding per-step gradients or Hessians and thereby achieving true batch scalability and substantial efficiency gains at scale.**

These clarifications demonstrate that the method is general, principled, and batch-scalable, rather than a narrow or incremental contribution. We are withdrawing to further strengthen the presentation and incorporate these clarifications more fully.

We appreciate the time and effort invested by the reviewers and committee.

Sincerely,

The Authors

**Withdrawal Confirmation:**

I have read and agree with the venue's withdrawal policy on behalf of myself and my co-authors.